# Neuronal sequences during theta rely on behavior-dependent spatial maps

**Eloy Parra-Barrero[1,2], Kamran Diba[3], Sen Cheng[1,2]\***

[1]Institute for Neural Computation, Ruhr University Bochum, Bochum, Germany; [2]International Graduate School of Neuroscience, Ruhr University Bochum, Bochum, Germany; [3]Department of Anesthesiology, University of Michigan, Michigan Medicine, Ann Arbor, United States

**Abstract** Navigation through space involves learning and representing relationships between past, current, and future locations. In mammals, this might rely on the hippocampal theta phase code, where in each cycle of the theta oscillation, spatial representations provided by neuronal sequences start behind the animal's true location and then sweep forward. However, the exact relationship between theta phase, represented position and true location remains unclear and even paradoxical. Here, we formalize previous notions of 'spatial' or 'temporal' theta sweeps that have appeared in the literature. We analyze single-cell and population variables in unit recordings from rat CA1 place cells and compare them to model simulations based on each of these schemes. We show that neither spatial nor temporal sweeps quantitatively accounts for how all relevant variables change with running speed. To reconcile these schemes with our observations, we introduce 'behavior-dependent' sweeps, in which theta sweep length and place field properties, such as size and phase precession, vary across the environment depending on the running speed characteristic of each location. These behavior-dependent spatial maps provide a structured heterogeneity that is essential for understanding the hippocampal code.

**\*For correspondence:**
sen.cheng@rub.de

**Competing interests:** The authors declare that no competing interests exist.

## Introduction

Hippocampal place cells elevate their firing rates within circumscribed regions in the environment ('place fields') (*O'Keefe and Dostrovsky, 1971*). In addition to this firing rate code, place cells display phase coding relative to the ~8 Hz theta oscillations of the hippocampal local field potential (LFP) (*Vanderwolf, 1969*). As animals cross a cell's place field, the cell fires at progressively earlier phases of the oscillation, a phenomenon known as theta phase precession (*O'Keefe and Recce, 1993*). At the population level, phase precession yields distinct neuronal sequences that traverse multiple positions within each cycle of the theta oscillation. The first cells to fire typically have place fields centered behind the current position of the animal, followed by cells with place fields centered progressively ahead, generating so-called 'theta sequences' (*Figure 1—figure supplement 1*; *Skaggs et al., 1996*; *Dragoi and Buzsáki, 2006*; *Foster and Wilson, 2007*). Formally, the position $r(t)$ represented by the hippocampal population at time $t$ appears to deviate from the physical location $x(t)$ of the animal, sweeping from past to future positions during each theta cycle (*Maurer et al., 2012*; *Gupta et al., 2012*; *Muessig et al., 2019*; *Kay et al., 2020*; *Zheng et al., 2021*). This phenomenon suggests that beyond the representation of the current spatial location of an animal, the hippocampal theta code more generally encompasses representations of past, present, and future events (*Dragoi and Buzsáki, 2006*; *Cei et al., 2014*; *Terada et al., 2017*). In support of this, theta phase precession and theta sequences have been reported to respond to elapsed time (*Pastalkova et al., 2008*; *Shimbo et al., 2021*), particular events and behaviors including jumping (*Lenck-Santini et al., 2008*), pulling a lever, and sampling an object (*Terada et al., 2017*; *Aronov et al., 2017*; *Robinson et al., 2017*). Phase-precession and theta sequences have also been

observed in brain regions other than the hippocampus (*Kim et al., 2012*; *Hafting et al., 2008*; *Jones and Wilson, 2005*; *van der Meer and Redish, 2011*; *Tingley et al., 2018*; *Tang et al., 2021*). These studies indicate that the theta phase code plays a supporting role in a wide array of cognitive functions, such as sequential learning (*Lisman and Idiart, 1995*; *Skaggs et al., 1996*; *Reifenstein et al., 2021*), prediction (*Lisman and Redish, 2009*; *Kay et al., 2020*), and planning (*Johnson and Redish, 2007*; *Erdem and Hasselmo, 2012*; *Bolding et al., 2020*; *Bush et al., 2015*).

However, for the hippocampal theta phase code to be useful, there must be a consistent relationship between theta phase, represented position $r(t)$, and the animal's true location $x(t)$. Surprisingly, this relationship has not previously been made explicit. It is often stated interchangeably that activity at certain phases of theta reflect positions 'behind' or 'ahead' of the animal, or, alternatively, in its 'past' or 'future', but these statements hint at two fundamentally different coding schemes. The first notion is that different theta phases encode positions at certain distances behind or ahead of the animal's current location. For example, within each theta cycle, the represented position $r(t)$ could sweep from the position two meters behind the current location to the position two meters ahead. The hippocampus would thus represent positions shifted *in space*: we therefore refer to this coding scheme as the 'spatial sweep' model. The second notion is that the hippocampus represents positions that were or will be reached at some *time* intervals into the past or future, respectively. For example, $r(t)$ might start in the position that was reached 5 s ago and extend to the position projected 5 s into the future. We call this the 'temporal sweep' scheme. Notably, the temporal sweep predicts different distances depending on speed of movement. In such a sweep, walking down a hallway, the prospective position reached in 5 s might be a couple of meters ahead, whereas when driving on the highway that position could be more than a hundred meters away. Thus, in temporal sweeps, the look-behind and look-ahead distances adapt to the speed of travel, which could make for a more efficient code.

In the following, we set out to develop a quantitative description of the hippocampal theta phase code by formalizing the spatial and temporal sweep models and comparing them to experimental results. Intriguingly, different experimental results appear to support different models. One the one hand, the length of theta sweeps has been shown to increase proportionally with running speed (*Maurer et al., 2012*; *Gupta et al., 2012*), a result which is consistent with temporal sweeps. On the other hand, place field parameters, such as place field size or the slope of the spikes' phase vs. position relationship (phase precession slope), do not change with speed (*Huxter et al., 2003*; *Maurer et al., 2012*; *Schmidt et al., 2009*; *Geisler et al., 2007*), which is consistent with spatial sweeps. To reconcile these apparently contradictory observations, we propose an additional coding scheme: the 'behavior-dependent sweep' based on a behavior-dependent spatial map that varies according to the animal's characteristic running speed at each location. We analyze recordings from rat CA1 place cells and in the same dataset reproduce the paradoxical results from different studies. Notably, we find that place fields are larger and have shallower phase precession slopes at locations where animals typically run faster, in support of the behavior-dependent sweep model. Finally, we compare simulated data generated from the three models based on experimentally measured trajectories and theta oscillations, and confirm that the behavior-dependent sweep uniquely accounts for the combination of experimental findings at the population and single-cell levels.

## Results

### Formalizing the theta phase code

We first put forward a mathematical framework for quantitatively describing the relationship between the current location of the animal, $x(t)$, and the position represented in the hippocampus at each moment in time, $r(t)$. While this framework considers the population representation, i.e. neuronal sequences during theta, the primary variable to be explained, it is linked to the activity of single place cells as follows. Each cell is considered to posses a 'true place field' (*Sanders et al., 2015*; *Lisman and Grace, 2005*), which corresponds to the positions represented by the cell, that is, a cell becomes active when the position represented by the population, $r(t)$, falls within the cell's true place field. Note that due to the prospective and retrospective aspects of the theta phase code, $r(t)$ reaches behind and ahead of $x(t)$, so a place cell will also fire when the animal is located ahead or behind its true place field. This results in a measured place field that is larger than the underlying

true place field. To stay consistent with the literature on place cells, in the following, when we refer to 'place field', we mean the measured place field.

First, we consider the spatial sweep scheme where the theta phase of a place cell's spike reflects the distance traveled through the place field (*Huxter et al., 2003*; *Geisler et al., 2007*; *Cei et al., 2014*) or equivalently, some measure of the distance to the field's preferred position (*Jeewajee et al., 2014*; *Huxter et al., 2008*; *Drieu and Zugaro, 2019*). In a uniform population of such phase coding neurons, the represented position, $r(t)$, sweeps forward within each theta cycle, starting at some distance behind the current location of the animal, $x(t)$, and ending at some distance ahead. We formalize these spatial sweeps by the equation:

$$r(t) = x(t) + d_\theta \frac{\theta(t) - \theta_0}{360},$$ (1)

where $d_\theta$, which we call the theta distance, is a parameter that determines the extent of the spatial sweep, $\theta(t)$ is the instantaneous theta phase in degrees, and $\theta_0$ is the theta phase at which the population represents the actual position of the animal. The value of $\theta_0$ effectively determines the proportion of the theta cycle that lies behind vs ahead of the animal. We note that none of our results or analyses depends on the particular value of this variable.

At the single-cell level, the spatial sweep model predicts that the phases of a cell's spikes will advance over a spatial range of $d_\theta$. Assuming that the phase advances over the whole theta cycle, the slope of spatial phase precession will be given by $m = -360°/d_\theta$ (*Equation 13* in Materials and methods). Place field sizes will be given by $s = s_0 + d_\theta$, where $s_0$ is the size of the true place field (*Equation 14* in Materials and methods). Both phase precession slopes and place field sizes are independent of running speed (*Figure 1B*), consistent with experimental observations (*Huxter et al., 2003*; *Schmidt et al., 2009*; *Geisler et al., 2007*; Sup. Fig. 1 in *Maurer et al., 2012*). At the population level, we can define the theta trajectory length as the difference between the maximum and minimum positions that $r(t)$ represents within a theta cycle. To a first-order approximation, that is, assuming that effects from acceleration and higher order derivatives are negligible, the theta trajectory length is given by $L = vT + d_\theta$, where $v$ is the running speed, and $T$ is the duration of a theta cycle (*Equation 11* in Materials and methods). While the spatial sweep predicts that theta trajectory length increases with running speed, the increase is small since the term $vT$, which represents the change in $x(t)$ due to the animal's motion during the theta cycle (*Figure 1A*, see also Figure 1 in *Maurer et al., 2012*) is smaller than $d_\theta$ for usual running speeds. Contrary to this prediction, experimental results show that the theta trajectory length is roughly proportional to running speed (Figure 2 in *Maurer et al., 2012*; Figure 1C in *Gupta et al., 2012* for speeds above 5 cm/s).

An alternative to the above is the temporal sweep model, where different phases of theta represent the positions that the animals have reached or will reach at some time interval in the past and future, respectively (see *Itskov et al., 2008* for a similar model). More formally,

$$r(t) = x\left(t + \tau_\theta \frac{\theta(t) - \theta_0}{360}\right),$$ (2)

where $\tau_\theta$ stands for the extent of the temporal sweep. To a first-order approximation, we derived for the theta trajectory length $L = (\tau_\theta + T)v$, which is proportional to running speed (*Equation 18* in Materials and methods) (*Figure 1C*). This is almost exactly the result reported by *Maurer et al., 2012*; *Gupta et al., 2012*. However, at the single-cell level, the temporal sweep model predicts that place field sizes increase and phase precession slopes become shallower with running speed (*Figure 1D*) – in clear conflict with the experimental results mentioned above. In particular, to a first-order approximation, theta phase precesses over the spatial range of $v\tau_\theta$. Thus, the slope of phase precession is given by $m = -360°/(v\tau_\theta)$, which increases hyperbolically with running speed (*Equation 19* in Materials and methods). Place field sizes would in turn increase linearly with speed, determined by $s = s_0 + v\tau_\theta$ (*Equation 20* in Materials and methods).

These conflicting results could potentially arise from differences in analytical methods (e.g. definition of key parameters, comparisons within versus across cells) or experimental procedures (e.g. task difficulty due to the use of linear vs. circular or complex mazes). If this was the case, spatial or temporal sweeps could adequately describe the theta phase code, perhaps switching dynamically

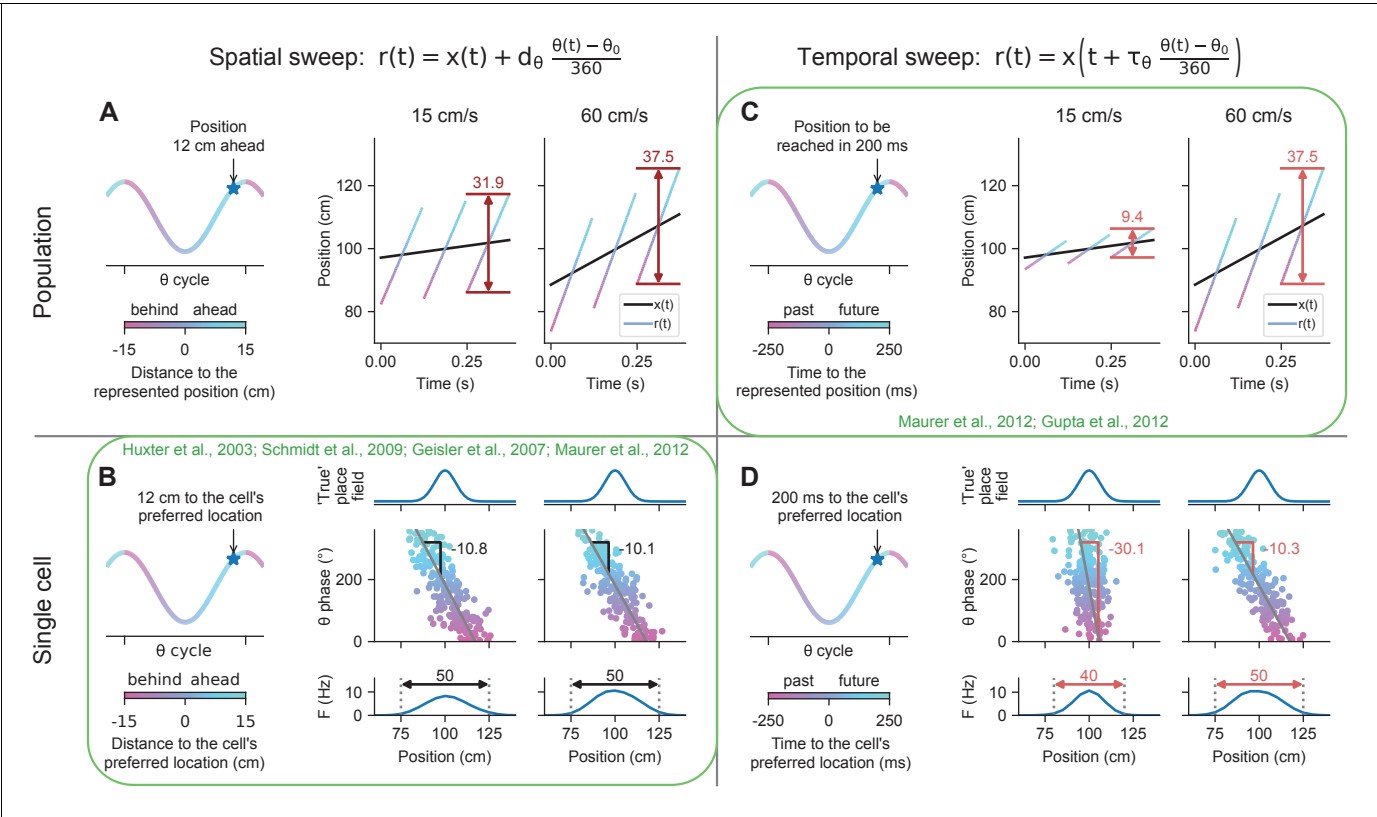

**Figure 1.** Simulated effect of running speed on population and single-cell properties in spatial sweep and temporal sweep models. Certain findings in previous studies (green) paradoxically support the spatial sweep at the single-cell level, but the temporal sweep at the population level. (**A**) Left. At different phases of theta, the population represents positions shifted behind or ahead in space by *fixed distances*. Right. The black lines represents the rat's actual location $x(t)$ as it runs through a linear track; the color-coded lines indicate theta trajectories represented by the place cell population $r(t)$. Since each theta trajectory starts and ends at fixed distances behind and ahead of the animal's current location, the length of a theta trajectory increases slightly with running speed (37.5 vs. 31.9 cm) to account for the animal's motion during the span of the theta cycle. (**B**) Left. At the single-cell level, the phases at which a cell spikes reflect the *distances* to the cell's preferred location. Right. The cell's preferred location is defined by its underlying 'true' place field (top). The cell fires proportionally to the activation of its true place field at $r(t)$, generating a phase precession cloud (middle) and corresponding measured place field (bottom). Phase precession slopes and place field sizes remain constant with running speed since, e. g., the cell always starts firing at 12 cm from the cell's preferred location. (**C**) Left. At different phases of theta, the population represents the positions that were or will be reached at *fixed time intervals* into the past or future, respectively. Right. A higher running speed leads to a proportionally increased theta trajectory length since, e.g., the position that will be reached in 200 ms is further ahead in space at higher speeds. (**D**) Left. At the single-cell level, the phase of theta reflects the *time* to reach the cell's preferred location. Right. At higher speed, the phase precession slope becomes shallower (−10.3 vs. −30.1 °/cm) and the size of the measured place field increases (50 vs. 40 cm) since, e.g., the cell will start signaling arrival at the cell's preferred location in 200 ms from an earlier position in space.

The online version of this article includes the following figure supplement(s) for figure 1:

**Figure supplement 1.** Schematic illustration of the relationship between place fields, theta phase precession, and theta sequences.

depending on the circumstances. Therefore, in the subsequent section, we attempt to reproduce and reexamine these different observations in the same dataset.

However, we also propose an additional scheme that, as it turns out, can potentially reconcile these findings. This third option combines the spatial sweep's constant place field sizes and phase precession slopes with the temporal sweep's increase in theta trajectory length with running speed. The former property requires that sweeps are independent of running speed whereas the latter requires that they depend on it. While at first this appears to be a plain contradiction, a resolution is offered if sweeps are made independent of the animal's instantaneous running speed $v(t)$, but dependent on their 'characteristic running speed' at each position $\bar{v}(x(t))$. This characteristic speed could be calculated as as a moving average (e.g. cumulative, exponentially weighted, etc.) as long

as it evolves sufficiently slowly, so that it can filter out variation in $v(t)$ from run to run. We refer to this model as the behavior-dependent sweep:

$$r(t) = x(t) + \bar{v}(x(t))\tau_\theta \frac{\theta(t) - \theta_0}{360} \qquad (3)$$

According to this model, at different phases of theta, the place cell population represents the positions that would have been or would be reached at certain time intervals in the past or future, like in the temporal sweep, but under the assumption that the animal runs at the characteristic speed through each location, rather than at the instantaneous running speed (*Figure 2A,B*). To avoid confusion in the following, when we use the terms 'speed' or 'running speed' without further qualifiers, we always refer to the instantaneous running speed. In the behavioral-dependent scheme, to a first-order approximation the length of a theta trajectory is given by $L = vT + \bar{v}(x(t))\tau_\theta$ (*Equation 22* in Materials and methods). Thus, the more stereotyped the behavior of the animal is (i.e. $v(t) \approx \bar{v}(x(t))$), the more the behavior-dependent sweep model would share the temporal sweep model's feature that theta trajectory length increases proportionally with running speed. At the same time, the behavior-dependent sweep model is formally equivalent to the spatial sweep model (*Equation 1*) with a theta distance that depends on the characteristic running speed at each position:

$$d_\theta = \bar{v}(x(t))\tau_\theta. \qquad (4)$$

Therefore, place fields, being fixed in space, remain unaltered by the instantaneous running speed, similar to in the spatial sweep model (*Figure 2B*). However, in regions of higher characteristic running speed, place fields are characterized by larger sizes and shallower phase precession slopes.

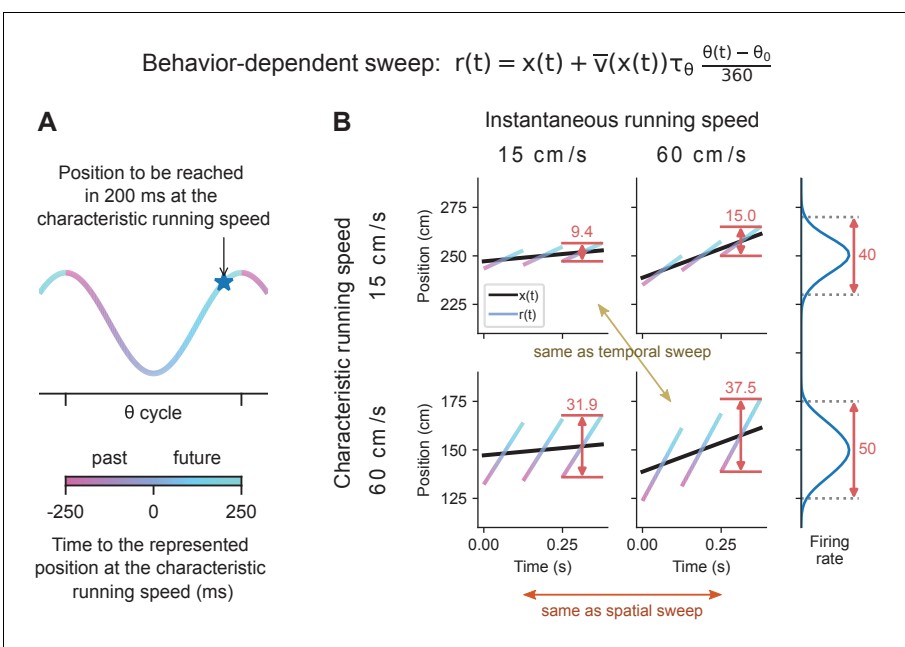

**Figure 2.** The behavior-dependent sweep integrates aspects of both spatial and temporal sweeps. Simulated data plotted as in *Figure 1A and C*. (A) Different phases of theta represent positions reached at different time intervals into the past or future assuming the animal ran at the characteristic speed at that location. (B) Comparison of theta trajectory lengths in areas of low and fast characteristic running speed (rows) at low and fast instantaneous running speeds (columns). When both characteristic and instantaneous running speeds coincide, the behavior-dependent sweep model and the first-order approximation of the temporal sweep model agree (ochre arrow). On the other hand, when changing the instantaneous speed at a given location, the behavior-dependent and spatial sweeps agree (orange arrow). Theta trajectory length is primarily determined by characteristic running speed. Instantaneous running speed has a modest effect caused by the larger change in $x(t)$ during the theta cycle at higher speeds. Place fields (right) are larger in areas of higher characteristic running speed, but do not change size with instantaneous running speed.

Specifically, the slope of phase precession is given by $m = -360°/(\bar{v}(x(t))\tau_\theta)$ (*Equation 23* in Materials and methods); and the place field size would be $s = s_0 + \bar{v}(x(t))\tau_\theta$ (*Equation 24* in Materials and methods).

## Theta trajectory lengths increase proportionally with running speed

To better evaluate the conflicting observations across the afore-mentioned studies, we next sought to reproduce their separate observations on the effect of running speed on population-level theta trajectories and single-neuron place field parameters within the same dataset and compare these against predictions from the three theta phase coding models.

To this end, we re-analyzed publicly available simultaneous single unit recordings from CA1 of 6 rats running on linear tracks for water rewards at both ends. Three of the sessions (*Mizuseki et al., 2013*) were from tracks that were familiar to the animal while the remaining three *Grosmark et al., 2016* were from novel tracks. To ensure that spatial representations and theta phase precession were stable for the entirety of the periods under examination, we discarded the first 5 min of the recording on the novel track, allowing sufficient time for hippocampal place fields and phase precession to have stabilized (*Frank et al., 2004*; *Cheng and Frank, 2008*).

First, we examined the effect of running speed on theta trajectory length. We performed a Bayesian decoding of positions from the ensemble neuronal firing (*Zhang et al., 1998*; *Davidson et al., 2009*) within each theta cycle in overlapping 90° bins with a step size of 30°. *Figure 3A* shows examples of individual decoded theta trajectories belonging to periods of slow (below 20 cm/s) and fast (above 40 cm/s) running in a sample experimental session. The lengths of the trajectories were

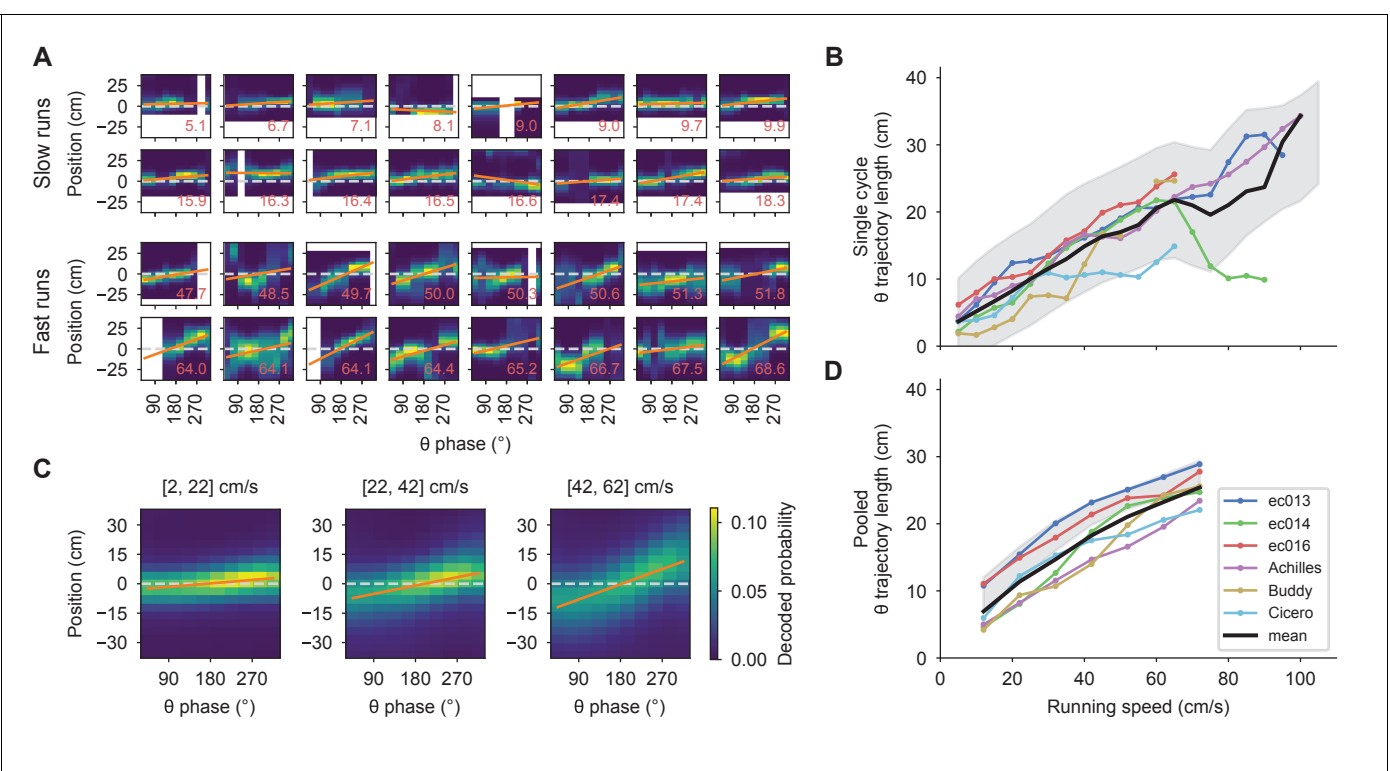

**Figure 3.** Theta trajectory lengths increase proportionally with running speed. (A) A random sample of theta sequences from a representative experimental session, with position probabilities decoded from the population spikes. Zero corresponds to the actual position of the rat at the middle of the theta cycle; negative positions are behind and positive ones, ahead. Orange lines indicate linear fits, from which theta trajectory lengths are computed. White pixels correspond to positions outside of the track or phase bins with no spikes. The red number at the lower right corner indicates the running speed in cm/s. (B) Moving averages for theta trajectory length for each animal using a 10 cm/s wide sliding window (colored lines). These averages are averaged again (thick black line) to obtain a grand average that weighs all animals equally. Shaded region indicates a standard deviation around the mean of all data points pooled across animals, which weigh each theta sequence equally. This mean does not necessarily match the grand average since different animals contribute different numbers of theta sequences. (C) Similar to A, but averaging the decoded probabilities across theta cycles belonging to the same speed bin indicated above the panel. (D) Similar to B for the averaged cycles, confirming the observation in B.

measured from best-fit lines. Despite the high level of variability, it was clear in the individual rats' averages and in the grand average across animals that the trajectory lengths increased roughly proportionately to running speed (*Figure 3B*). We saw the same effect when we averaged the decoded position probabilities over all cycles belonging to the same speed bin before fitting a line and calculating the theta trajectory length (*Figure 3C,D*). For this analysis, we defined seven 20 cm/s wide overlapping speed bins starting at 2 cm/s in increments of 10 cm/s. Thus, our results on the dependence of theta trajectory lengths on running speed were consistent with those of *Maurer et al., 2012* and *Gupta et al., 2012*, and appear to be supportive of the temporal sweep model. Alternatively, the results could be accounted for by the behavior-dependent sweep model, if running behavior was sufficiently stereotyped across the track, a question to which we return below.

## Individual place field sizes are independent of running speed

We next analyzed the effect of running speed on place field size. First, we calculated firing rates separately for periods with different running speeds using the same speed bins as before. As the number of trials in each spatial bin at a given running speeds were limited, we calculated place field sizes only when a field was sufficiently well sampled at a given speed (see Materials and methods). Visual inspection revealed that individual fields tended to maintain similar sizes regardless of running speed (*Figure 4A*). To quantify the relationship more systematically, we performed linear regression on place field size vs. speed for each field and examined the slopes of the best fitting lines (*Figure 4A*, bottom row). *Figure 4B* shows that for each rat, slopes variably took on both negative and positive values, with no clear bias towards either ($p>0.3$, Wilcoxon signed-rank test). Thus, we found no evidence for a systematic increase in place field size with running speed.

Intriguingly, however, when we combined all place field sizes corresponding to each speed bin, we uncovered a linear relationship between place-field size and speed (*Figure 4C*; $p<0.02$ for the association between place field sizes combined across fields and speed bin for five of the six animals, Kendall's $\tau$ test). This increase in place field sizes is consistent with the temporal sweep model, whereas the lack of within-field changes mentioned above supports spatial sweeps. Both findings, however, can be consistent with the behavior-dependent sweep model provided that place fields are larger in track areas characterized by faster running (*Figure 2*). We examine this question below.

## Phase precession slopes of individual place fields are independent of running speed

Similarly conflicting results were obtained when analyzing phase precession. We calculated phase precession slopes at each running speed bin for each field separately by pooling spikes across different field traversals according to the instantaneous running speed of the animal when the spikes were emitted. *Figure 5A* shows the results of this analysis for three example place fields. To ensure that we capture and adequately describe steep phase precession in small place fields (e.g. *Figure 1D*), we devised a method using orthogonal regression (see Materials and methods). An analysis of all place fields (*Figure 5B*) revealed that phase precession slopes of individual fields did not change systematically with running speed for any of the six animals ($p>0.05$, Wilcoxon signed-rank test). Remarkably, however, when we combined phase precession slopes across different fields for each speed bin, the combined slopes at slower speeds tended to be steeper than at faster speeds (*Figure 5C*; $p<0.006$ for the association between phase precession slopes combined across fields and speed for five of the six animals, Kendall's $\tau$ test).

In this analysis, when rats accelerate or decelerate through a field, different parts of the field's phase precession cloud are sampled at different running speeds. This can be seen in the first field (no. 4) in *Figure 5A*. Since the rat was accelerating through the field, spikes emitted at lower and higher speeds appeared more frequently at the beginning and end of the field, respectively. Phase precession clouds have some curvature (*Souza and Tort, 2017*; *Yamaguchi et al., 2002*), so sampling different parts of the field under acceleration or deceleration could lead to a spurious relationship between speed and phase precession slope. To avoid this confound, we further analyzed phase precession slopes in single passes through a field (*Schmidt et al., 2009*). We only included passes that occurred at approximately constant speed, that is, passes with a coefficient of variation in speed lower than 0.3. *Figure 5D* shows examples of individual passes through two fields, sorted by running speed. Note how, notwithstanding variability in the slopes, there is no systematic change with

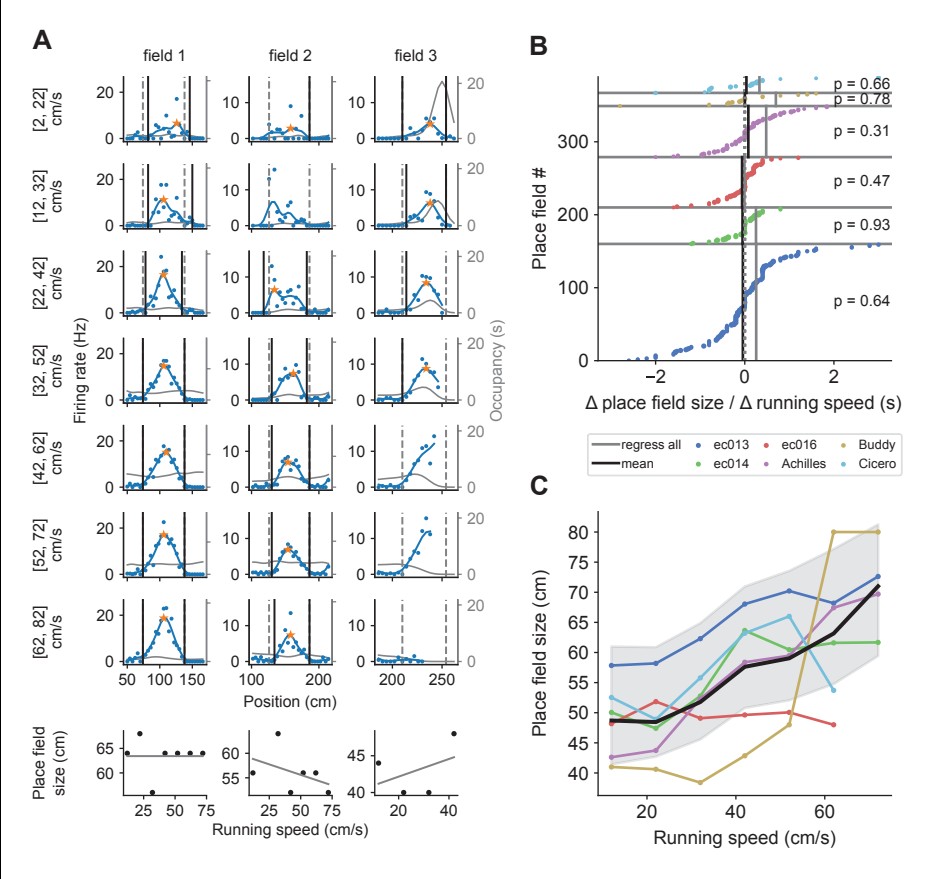

**Figure 4.** Place field size increases with running speed when combining data across fields, but not for individual fields. (A) The size of a given place field remains roughly constant regardless of running speed in examples from three individual place fields (one per column). Dashed gray lines represent the extent of the place fields calculated from the complete set of spikes at all running speeds. Black lines mark the extent of the place field calculated for each speed bin. Where only one end of the place field could be determined (e.g., field 3, third and fourth rows), place field size was set to twice the distance from the field's peak (orange star) to the detected field end (see Materials and methods). Thin gray lines represent the occupancy (time spent) per spatial bin (axis on the right). At the bottom, linear regressions on place field size versus running speed for each place field. Fields 2 and 3 belong to the same cell. (B) The slopes from linear regression of place field size vs. running speed for all place fields, sorted for each animal. Across the population of place fields, slopes were not significantly different from zero (indicated p values). The size of the dot reflects the number of data points that contributed to the regression. The black vertical lines indicate the weighted averages of these slopes for each animal. The gray lines indicate the slope of the regression calculated by first pooling together data points from all place fields for each animal. (C) Remarkably, when combining data across fields, the average field size generally increases as a function of running speed. Colored lines represent individual animals, and the thick black line averages over them. Shading represents the standard deviation around the mean of all data points pooled across animals.

The online version of this article includes the following figure supplement(s) for figure 4:

**Figure supplement 1.** Restricting the theta trajectory length analysis to areas covered by the place fields analyzed does not change the results meaningfully.

running speed, even though the speeds nearly double. There was no significant relationship between phase precession slope and speed for individual fields in any of the three animals with sufficient data for this analysis, that is, more than 10 fields (*Figure 5E*). Yet again, when we combined phase precession slopes across place fields, we saw that they become shallower with speed ($p<0.04$ for five of the six animals, Kendall's $\tau$ test; *Figure 5F*). Hence, the results from single pass phase precession analysis confirm those obtained from the session-wide analysis.

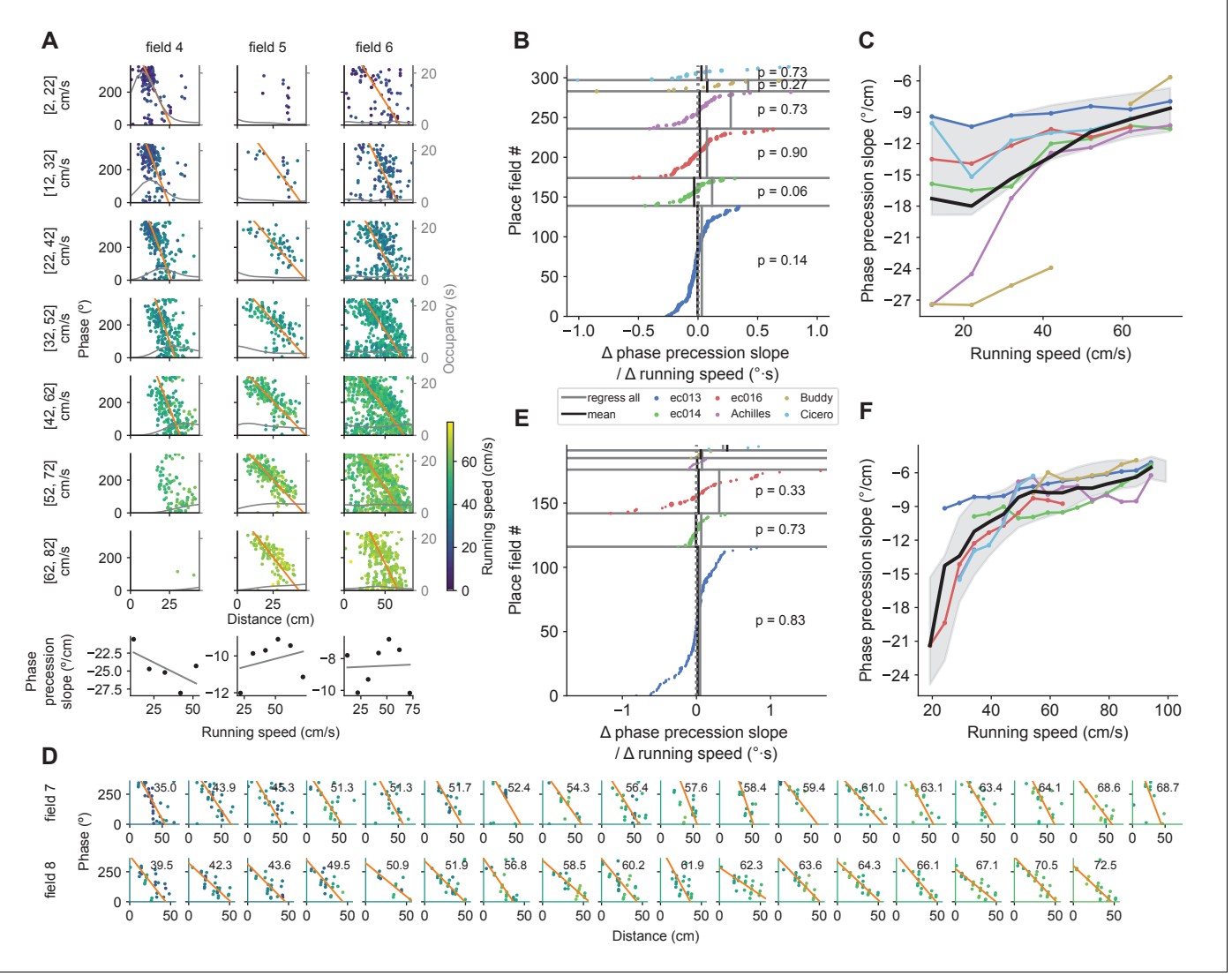

**Figure 5.** Phase precession slope increases with running speed when combining data across fields, but not for individual fields. (A) Example phase precession slopes at different speeds for three fields. Instantaneous speeds when the spikes where emitted are color coded. Thin gray line displays the occupancy. At the bottom, linear regressions on phase precession slope versus running speed for each place field. (B) Similar to *Figure 4B* for the slopes of the linear regressions on phase precession slope vs. running speed for individual fields. (C) Similar to *Figure 4C* for phase precession slopes combined across fields for each speed bin. (D) Example phase precessions in single passes through two place fields (rows) sorted by running speed (upper right corner, in cm/s). Color code as in A. (E, F) Same as in B and C but for single-trial phase precession slopes. The means for each animal in F were calculated as a moving average on a 10 cm/s wide sliding window.

## Theta trajectories and place field parameters vary according to characteristic running speed across the track

We have shown that theta trajectory lengths and place field sizes increase with running speed, and phase precession slopes become shallower, when combining data-points from different positions along the track or place fields (*Figure 3B,D*; *Figure 4C* and *Figure 5C,F*), but not when fields are analyzed individually (*Figure 4B* and *Figure 5B,E*). While these combined findings appear counter-intuitive, the behavior-dependent sweep model offers a simple explanation: short theta trajectories and small place field sizes could occur at locations where animals tend to run slowly, contributing to the low-speed averages, while long theta trajectories and large place field sizes could occur where animals tend to run fast, contributing to the fast-speed averages (*Figure 6A*). For this explanation to be valid, the following prerequisites must hold: (1) running speeds should differ systematically along the track such that (2) place fields in different positions contribute differently to the low- and high-

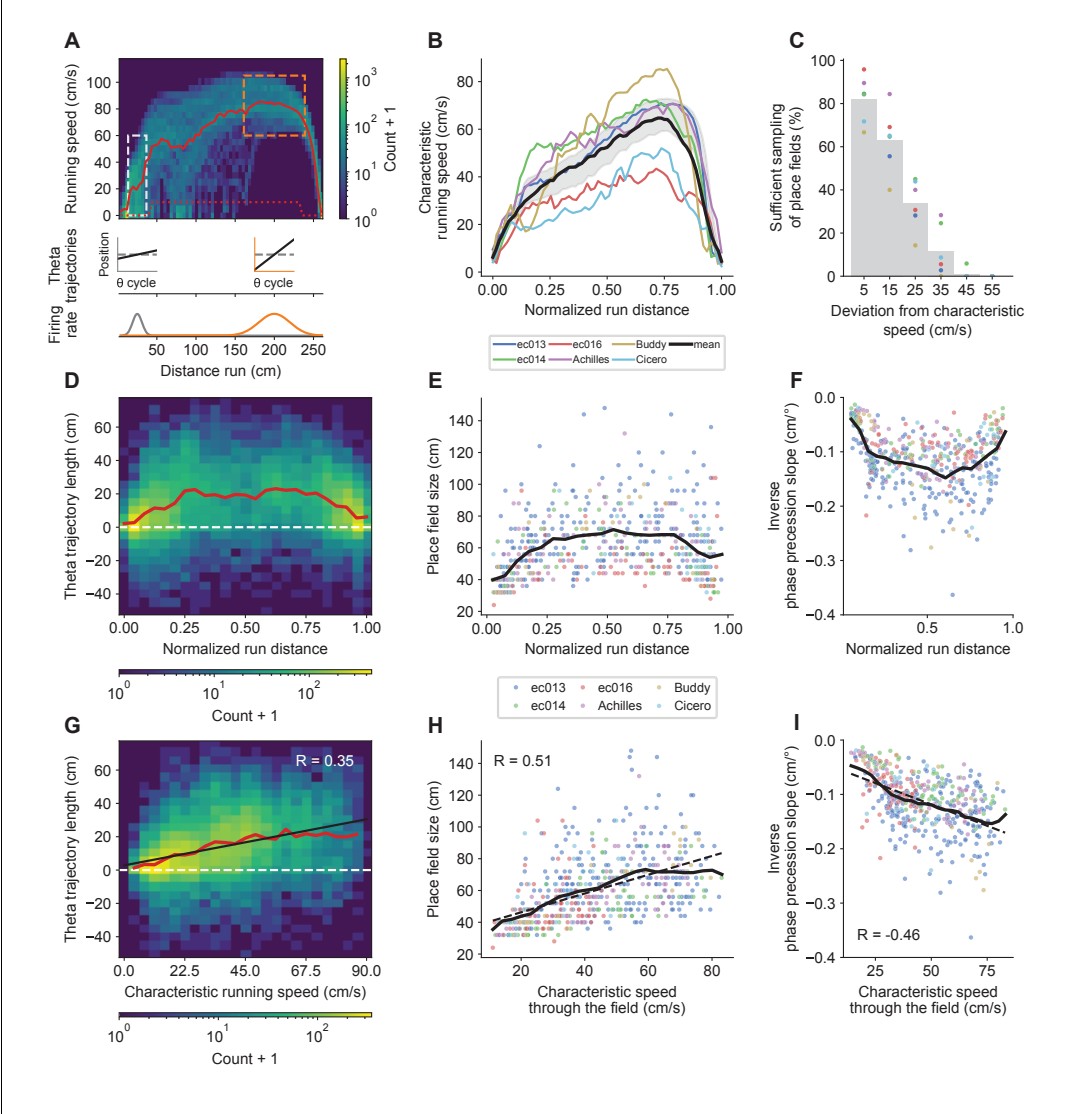

**Figure 6.** Structured place field and theta trajectory heterogeneity correlates with characteristic speed. (**A**) Potential explanation for the increase in average theta trajectory length, place field size and phase precession slopes with speed despite the lack of systematic within-field changes. The histogram shows the distribution of running speeds by positions for rightward runs in one experimental session. The thick red line is the characteristic speed, defined as the mean speed after discarding trials with running speed < 10 cm/s in the center of the track (exclusion criteria indicated by dotted red line). Running speeds tend to cluster around the characteristic speed. (**B**) Average characteristic speed as a function of the normalized distance from the start of the run for each animal (colored lines) and the grand average across animals (thick black line). Shaded region as in *Figure 3B*. (**C**) The proportion of fields sufficiently sampled for place field analysis at a certain speed bin falls steeply with the deviation between the speed bin and the mean characteristic speed through the field. Grey bars indicate averages across animals. (**D**) Histogram of individual theta trajectory lengths across the track for all animals combined and their mean (red line). Negative values correspond to theta trajectories going in the opposite directional as running. (**E**) Place field sizes and (**F**) the inverse of phase precession slopes across the track for each animal (colored dots), and their moving average with a window size of 0.1 (normalized units; black lines). (**G**) Histogram of theta trajectory lengths (with mean [red line] and linear fit [black line]) versus characteristic running speed for locations where each trajectory was observed. (**H**) Place fields are larger and with (**I**) shallower phase precession slopes for fields with higher mean characteristic speed. Solid black lines represent moving averages with a window size of 15 cm/s, and dashed lines indicate linear fits.

The online version of this article includes the following figure supplement(s) for figure 6:

**Figure supplement 1.** Like D, G, separated by animal.

**Figure supplement 2.** Like E, F, H, and I, separated by animal.

**Figure supplement 3.** Linear relationships between place field sizes, phase precession slopes, and theta trajectory lengths.

speed analyses; and (3) theta trajectory lengths, place field sizes and the inverse of phase precession slopes depend linearly on the characteristic speed across different locations (*Equation 22-24*).

To test the first prerequisite, we calculated the characteristic speed for each spatial bin along the track. This was defined as the mean speed after discarding trials where the animal stopped running on the track (speeds < 10 cm/s), to prevent atypical pauses from distorting the mean values. The upper panel in *Figure 6A* shows the characteristic speed for rightward runs in one sample experimental session and *Figure 6B* shows averages across running directions and sessions. All animals showed similar asymmetric inverted U-shape relationships between characteristic speed and position along the track. Next, as required by the second prerequisite, we confirmed that running speeds at each position were clustered around the characteristic running speed (*Figure 6C*). We also observed inverted U-shape relationships between position along the track, and theta trajectory lengths and place field sizes (*Figure 6D,E*). Because theta trajectory lengths and place field sizes should be linearly related to the inverse of phase precession slopes (*Equation 22-24*), we plotted this variable along the track and found the corresponding U-shape distribution (*Figure 6F*). Finally, in support of the third prerequisite, we found that theta trajectory lengths and place field sizes increased, and inverse phase precession slopes decreased, roughly in proportion to characteristic speed for up to at least 60 cm/s, which encompassed most of the data (*Figure 6G–I*). Some flattening of the curves appeared after 60 cm/s, but this was less apparent when considering each animal individually (*Figure 6—figure supplement 1* and *Figure 6—figure supplement 2*), suggesting that the flattening is at least partly caused by combining points from clouds with different slopes and speed ranges.

Since theta trajectory lengths, place field sizes and inverse phase precession slopes all displayed matching distributions, we tested their mutual relationships. We found that the three of them displayed strikingly linear relationships to one another (*Figure 6—figure supplement 3*), underscoring the intimate relationships that hold between these systematically interrelated measures of place cell activity. It is also worth noting that theta trajectory length correlates slightly better with characteristic running speed than with either mean place field size or mean inverse phase precession slope at the locations where the trajectories occurred (R = 0.35 vs 0.29 for the latter two; *Figure 6—figure supplement 3B,C*).

Since speed and location on the track are correlated, we also tested whether place field parameters could be best explained by distance to the nearest end of the track, rather than by characteristic running speed. We performed a likelihood ratio test that indicated that both variables contributed significantly to the prediction of place field sizes and inverse phase precession slopes ($p<10^{-6}$). However, the fraction of the total variance accounted for differed substantially between the two variables. A model combining both variables could account for 29% and 28% of the variance in place field sizes and inverse phase precession slopes, respectively. Out of this, characteristic running speed uniquely accounted for 13% and 11% of the variance, whereas distance to the border uniquely accounted for 3% and 7%, respectively. Therefore we conclude that although both variables are related and contribute to determining place field parameters, characteristic running speed plays a more prominent role.

Taken together, these findings account for and reconcile the speed-dependence of average theta trajectory lengths, place field sizes and phase precession slopes with the speed-independence of individual fields, and satisfy the prerequisites of the behavior-dependent sweep model.

## The behavior-dependent sweep model accounts best for all of these experimental results

Our analyses so far suggest that neither the spatial nor the temporal sweep models can account for all of the experimental observations, whereas the behavior-dependent sweep appears consistent with all of them. However, these analyses might involve too many simplifying assumptions about how running speeds, theta coupling and place field parameters are co-distributed. To address this issue, we developed generative models based on the experimental LFP and animal tracking data to generate spikes according to each of the different theta phase coding schemes. The parameters in each model were adjusted to account for as many features in the experimental data as possible. We analyzed the effect of instantaneous running speed on theta trajectory lengths, place field sizes and phase precession slopes in each of the simulated datasets alongside the experimental data (*Figure 7*). This approach further allowed us to examine whether any of the models could provide a simple unified account of the data, using a single set of parameters.

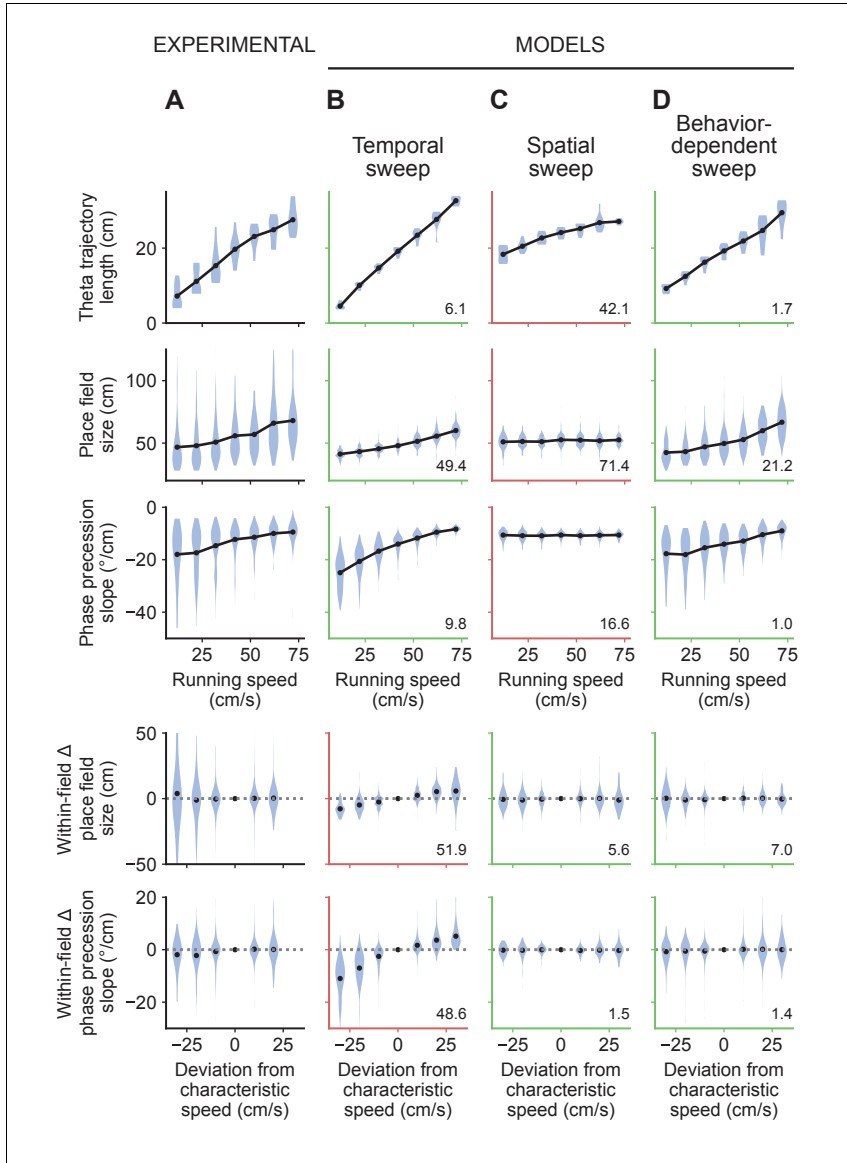

**Figure 7.** Only the behavior-dependent sweep model accounts for all experimental observations. (**A**) Summary of experimental results from *Figures 3–5*. Results from all animals are combined. Only one animal featured more than three sessions. For this animal, we sub-selected three sessions at random so that it would not dominate the results. Top three rows: theta trajectory lengths, place field sizes and phase precession slopes combined across positions and fields based on instantaneous running speed. Bottom two rows: the changes in individual place fields' sizes and phase precession slopes as a function of the difference between instantaneous running speed and the characteristic running speed. For this analysis, fields were assigned to the speed bins closest to the characteristic running speeds at their locations. Black dots represent average values. (**B**) Similar to A for data generated by the temporal sweep model. The temporal sweep model provides a qualitative fit to the average theta trajectory lengths, place field sizes and phase precession slopes, but incorrectly predicts increases in individual place field sizes and phase precession slopes. In this and other columns, green and red axes indicate good and bad qualitative fits, respectively (i.e. whether the model predicts the same type of change in the variables with speed as the experimental data exhibits) and the numbers in the lower right corners indicate the mean squared differences between the average values produced by the model and the experimental data across speed bins. (**C**) The spatial sweep model accounts for the lack of within-field increases in place field sizes and phase precession slopes. However, the increase in theta trajectory length with running speed is not nearly large enough and place field sizes and phase precession slopes remain flat with running speed. (**D**) The behavior-dependent sweep model captures both the population average and within-field effects, providing a good agreement with all experimental results.

*Figure 7 continued on next page*

*Figure 7 continued*

The online version of this article includes the following figure supplement(s) for figure 7:

**Figure supplement 1.** The behavior-dependent sweep model captures changes in place field skewness with acceleration.

**Figure supplement 2.** The behavior-dependent sweep model captures changes in peak firing rates with speed.

The temporal sweep model roughly captures the observed increase in theta trajectory lengths, place field sizes and phase precession slopes combined across different positions and fields based on instantaneous running speed. However, it did so for the wrong reasons, since it incorrectly predicts changes in individual fields with speed (*Figure 7B*). By contrast, the spatial sweep model accounts for the speed-independence of individual fields, but severely undershoots the increase in theta trajectory length with running speed and fails to capture the increases in place field sizes and phase precession slopes (*Figure 7C*). Only the behavior-dependent sweep model is able to replicate the correct combination of qualitative effects (*Figure 7D*). Furthermore, the behavior-dependent sweep produced values with the lowest mean squared deviation to the experimental data for four of the five variables considered (*Figure 7*; numbers in corners).

Further proof for the validity of the behavior-dependent sweep model comes from the fact that we fitted one of its crucial parameters based on a different subset of the data as the one used to evaluate the models. In particular, we estimated $\tau_\theta$ based on *Equation 23* from the experimentally observed relationship between phase precession slopes and characteristic running speed (*Figure 6I*), yielding $\tau_\theta = 0.57$ s. With this value, the model provided a good fit to the relationship between instantaneous running speed and theta trajectory lengths, place field sizes and phase precession slopes (*Figure 7D*). This highlights the ability of the model to account for the relationships and internal consistency of multiple measurements of place cell activity.

Finally, a subtle prediction of the behavior-dependent sweep model is that fields will skew differently depending on whether the animal is typically accelerating or decelerating through the field–that is when the characteristic running speed changes within the field. We therefore analyzed the relationship between mean acceleration and place field skew in the experimental data and found a good qualitative fit with the predictions from the model (*Figure 7—figure supplement 1*). Negatively skewed fields *Mehta et al., 2000* were observed at all acceleration values, but positively skewed fields occurred mainly at positive accelerations, resulting overall in a significant correlation between mean acceleration and skew ($p = 4 \times 10^{-8}$, Kendall's $\tau$ test). A similar observation was made by *Diba and Buzsáki, 2008*, who reported positive skews at the beginning of the track, where rats are usually accelerating, and negative skews towards the end, where rats decelerate.

In summary, we found that the behavior-dependent sweep model provided the best overall fit to the experimental data with a single set of model parameters.

## Discussion

While research over the past four decades has shed light on numerous aspects of the hippocampal code, the exact relationship between represented position, theta phase, and the animal's physical location has remained ambiguous. The expression of this relationship in theta sequence trajectories at the population level and in theta phase precession at the single-cell level are two sides of the same coin, but the two levels have seldom been explored side by side. Here, we illustrated that established findings in the field at these different levels are in apparent contradiction with one another. We reconciled these findings by demonstrating that even though individual place fields do not change with instantaneous running speed, their size and phase precession slopes vary according to the animal's characteristic running speed through these place fields. In particular, regions of slow running are tiled with smaller place fields with steeper phase precession, corresponding to shorter theta trajectories at the population level, whereas regions of fast running are covered by larger place fields with shallower phase precession, corresponding to longer theta trajectories. Based on these observations, we proposed a novel theta phase coding scheme, the behavior-dependent sweep. In this model, the spatial extent of the theta oscillation sweep varies across the environment in proportion to the characteristic running speed at each location. As a result, the sweeps can be seen as

going through the positions that were or will be reached at certain time intervals in the past or future, assuming the animal runs at the speed characteristic of each place, as opposed to at its actual instantaneous speed. This coding scheme thus integrates aspects of both the spatial and temporal sweeps.

## Relationship to previous studies on the speed-dependence of the hippocampal code

Why were the relationships between characteristic speed, hippocampal sweeps and place field parameters not evident in previous studies? Two previous studies (*Maurer et al., 2012*; *Gupta et al., 2012*) pooled theta trajectories across different positions when showing the effect of speed on theta trajectory length, therefore masking the role of position in mediating the relationship between the two variables. In other studies, the dependence of place field parameters on the characteristic running speed through the field did not emerge because analyses focused on relative measures between pairs of cells (*Geisler et al., 2007*; *Diba and Buzsáki, 2008*) or individual fields (*Schmidt et al., 2009*; *Huxter et al., 2003*; *Maurer et al., 2012*) that remain unaffected.

We note that one other study (*Chadwick et al., 2015*) previously proposed a model to account for the increase in theta trajectory length with running speed despite stable place field sizes and phase-precession. The model achieved this by increasing the precision of the theta phase versus position relationships of cells with running speed. This feature, however, results in unusually noisy phase-precession at low speeds (see Neuronal sequences during theta rely on behavior-dependent spatial maps for results on a model that operates based on the same principle and comparisons to experimental data). Furthermore, this kind of model cannot account for the dependence of average place field size on characteristic running speed.

On the other hand, two models which are consistent with the increase in place field sizes with characteristic speed did not address theta phase coding. Slow feature analysis attempts to extract slowly varying signals from the sensory input, and has been found to account for different properties of place, head direction and view cells (*Franzius et al., 2007*). Extracting place-specific features that vary on the order of several seconds, for instance, would lead to larger place fields at locations of higher speeds, as we have observed. This observation is also consistent with the successor representation model of place-fields (*Stachenfeld et al., 2017*). In this model, the activity of a cell is determined by the expected discounted future occupancy of the cell's preferred location, which proves useful for planning. Thus, a cell with a preferred location where average running speed is higher would begin firing relatively further away from that place. The behavior-dependent theta sweep model shares this prediction with these two models, and further accounts for theta-phase and trajectory relationships.

## Behavior-dependent sweeps are advantageous for prediction and planning

Given the purported role of theta phase coding in prediction and planning, we can compare our three coding schemes in terms of their benefit for these functions. Representing future positions based on the distance to them, as in the spatial sweep, could allow an animal to calculate the speed necessary to reach those positions in a given amount of time. This representation might also be convenient because our bodies and sensory organs limit the spatial range at which we can interact with the world. In contrast, looking forward based on an interval of time rather than space, as in the temporal sweep model, could increase the code's efficiency by matching the extent of the look-ahead to the speed of progression through the environment. For instance, while driving a car on a highway, we might have to look ahead by several hundred meters, since those positions are just seconds away, while such a distance is not useful when strolling in a park. Plans made too far in advance are very likely to be discarded upon unexpected changes or as new information comes to light, which is conceivably related to the use of a discount factor in reinforcement learning models (*Sutton and Barto, 1998*). Temporal sweeps also seem more plausible for phase-coding in non-spatial domains (*Pastalkova et al., 2008*; *Lenck-Santini et al., 2008*; *Robinson et al., 2017*; *Terada et al., 2017*; *Aronov et al., 2017*). However, temporal sweeps require a fine control by the instantaneous running speed, and there are indications that the hippocampal network lacks the the flexibility required to

modify the time lags between pairs of cells at the theta timescale (*Diba and Buzsáki, 2008*; *Shimbo et al., 2021*).

The behavior-dependent sweep might thus represent the best solution compatible with network constraints by taking advantage of the relative simplicity of spatial sweeps while still enjoying the benefits of temporal sweeps in situations in which behavior is stereotyped. Thus, the look ahead on a highway remains a few hundred meters even if on occasion we are stuck in a traffic jam. Finally, behavior-dependent sweeps could make predictions of future positions more robust through the averaging of past experience.

Despite the potential benefits of behavior-dependent sweeps mentioned here, these sweeps are probably best construed as a default or baseline mode of operation, which could be further modulated by ongoing cognitive demands and task contingencies (*Wikenheiser and Redish, 2015*; *Shimbo et al., 2021*; *Zheng et al., 2021*).

## Look-ahead times can help explain place field sizes

All theta sweep models and concepts of prospective and retrospective coding, and extra-field spikes in general, imply that place cells also fire outside their true place fields. In other words, an experimentally measured place field encompasses the extent of the true place field, as well as look-ahead and look-behind activity. This link between place fields and prospective/restrospective representations at the theta timescale provides a lower bound on place field sizes for a given task based on its characteristic running speed. For sweeps that are centered around the current position, a $\tau_\theta$ of $\sim 0.6$ s means that theta sweeps on average end in positions arrived at in $\sim 300$ ms. Representing those future positions would be of use only if there was enough time for the animal to modify its behavior in response to them. Reaction times for rats performing simple and choice tasks range between 150 ms and 500 ms (*Baunez et al., 2001*; *Brown and Robbins, 1991*), and stop signal reaction times stand at around 360 ms (*Eagle et al., 2009*). Therefore, place field sizes associated with a look-ahead time of 300 ms probably stand towards the lower end of what can be used to control behavior through rapid decision-making. However, place fields (*Kjelstrup et al., 2008*) increase in size along the dorsoventral axis of the hippocampus. The largest place fields recorded in rats stand at about 10 m, which would correspond to representing a time window of > 10 s. Thus, we can conceive of a hierarchy of subregions corresponding to predictions at different temporal scales.

## Potential mechanisms of behavior-dependent sweeps

The dependence of place field parameters and theta trajectories on characteristic speed raises the question of how and when this speed might be computed in the brain, and how it is used to control hippocampal sweeps. One intriguing possibility is that the animal can estimate the speed at which it will typically traverse some region of an environment based on general knowledge of similar environments (e.g. knowing that speed will tend to be lower around boundaries). Another possibility is that the characteristic speed is taken to be the speed of the firsts traversals through the environment, in which theta sequences are reportedly lacking or reduced (*Feng et al., 2015*; *Tang et al., 2021*), and then updated incrementally.

The latter suggestion fits naturally with network connectivity models of phase precession (*Jensen and Lisman, 1996*; *Tsodyks et al., 1996*; *Drieu and Zugaro, 2019*). In these models, theta sequences are produced by the propagation of activity through the population, perhaps owing to the recurrent connectivity in area CA3 (*Cheng, 2013*; *Azizi et al., 2013*). The network is cued with the current position and recalls upcoming positions. As the animal moves forward, positions will activate at earlier phases of the theta cycle, producing phase precession. In this family of models, it is easy to imagine sweeps varying based on average speed. Place cells could be connected with a higher synaptic strength the closer in time they become activated, leading to stronger and further reaching connectivity in areas through which the animals run faster on average. If the speed of propagation in the network is then made dependent on synaptic strength, activity would propagate further in the network within each theta cycle where speed had been higher, producing longer theta trajectories in those areas. Alternatively, based on the view that theta sequences arise from independently phase-precessing cells (*Chadwick et al., 2015*), behavior-dependent sweeps could simply result from a behavior-dependent spatial map that yields larger place fields in areas of faster running.

## Predictions and outlook

In the dataset analyzed, changes in running speed arose naturally from animals running on a finite track, but were not controlled by the experimenter. Further confirmation for the behavior-dependent sweep model could come from experiments in more complex environments that can produce variable running speeds at different specific locations (*Kropff et al., 2021*), such as slowing down in previously fast segments and speeding up in slow ones. This would allow for the comparison between behavior-dependent sweeps, and alternative models for which the look-behind and look-ahead distances vary depending on other variables. In particular, it would be interesting to dissociate the effect of characteristic running speed from that of distance to the environmental boundaries, since the boundary vector cell (*Barry et al., 2006*) and Laplace transform (*Sheehan et al., 2021*) models suggest that it could be some non-linear function of distance to the border or trial start, rather than characteristic running speed, that determines place field parameters. Other factors that could play a role are the distance to reward locations (*Hollup et al., 2001*), the distance to the goal (*Wikenheiser and Redish, 2015*), the density of sensory cues, the spatial rate of change of visual input (*Tanni et al., 2021*), the complexity of the maze, or the amount of retrospection and prospection required for the task.

Although we have focused primarily on the effect of speed, it is reasonable to also expect acceleration to have an effect on theta phase coding. We discuss three specific mechanisms. First, acceleration could introduce a second-order term in the equation for $r(t)$ such that theta trajectories curve in areas through which the animal typically accelerates or decelerates. This might cause trajectories in areas of positive acceleration to reach further ahead than behind, and trajectories in areas of negative acceleration to reach further behind than ahead, consistent with the findings of *Gupta et al., 2012* and *Bieri et al., 2014*. This effect would also be modulated by the phase at which the current position is represented ($\theta_0$ in our equations), since this controls the ratio of look-ahead to look-behind. For instance, the more look-ahead there is, the longer the theta trajectories would become in areas of positive acceleration. Second, acceleration leads to different characteristic running speeds in different parts of a place field. Our model predicts that this would cause place fields to skew differently in areas of positive and negative acceleration, a prediction for which we have found some preliminary experimental support (*Figure 7—figure supplement 1*). Additionally, the model indicates that changes in characteristic speeds through fields cause corresponding changes in theta phase precession. If characteristic speed increases across a field, the slope of phase precession would become increasingly shallow, resulting in concave phase precession clouds, whereas if characteristic speed decreases across the field, the slope of phase precession would be increasingly steep, producing convex 'banana shaped' phase precession clouds (*Souza and Tort, 2017*; *Yamaguchi et al., 2002*). Future studies are required to test these two predictions, since the range of accelerations in the current dataset was relatively small, especially for positive values, and strong acceleration and deceleration were concentrated over small regions at the start and end of the track, respectively. Third, theta frequency has recently been shown to increase with positive acceleration, but not deceleration or speed (*Kropff et al., 2021*). One might wonder whether shorter theta cycles due to positive acceleration, rather than the lower running speed, could explain the shorter theta trajectories we observed at the beginning of the track. However, we also observe shorter theta trajectories at the end of the track, where running speeds are also lower, but acceleration is negative and theta cycles are longest (*Figure 6B,C*). Furthermore, as shown by *Kropff et al., 2021*, the increase in theta frequency is accompanied by an increase in the intrinsic firing frequency of cells which would provide a compensatory effect on theta trajectories. Even without such compensation, the reduction in theta trajectory length due to shortened theta cycle is small and insufficient to account for the observed effect.

Lastly, our behavior-dependent sweep model could be applied to describing the activity of place cells in two-dimensional environments, and potentially also that of grid cells, since theta phase coding has been demonstrated in both of these cases (*Huxter et al., 2008*; *Jeewajee et al., 2014*; *Hafting et al., 2008*). For grid cells, our model predicts that sizes of individual firing fields within the grid will vary according to running speeds characteristic of the corresponding maze regions. By contrast, the distance between firing field peaks would remain constant, since the locations of the underlying true fields remain fixed in the model.

In conclusion, our analyses have highlighted the heretofore unresolved tension lurking in the field between experimental results and interpretations explicitly or implicitly supporting spatial and temporal sweeps. Our novel interpretation reconciles apparently contradictory findings by making spatial sweeps dependent on the average behavior at each position, combining features and benefits of both kinds of sweep. In doing so, the model advances our understanding of how the mammalian brain represents past, present and future spatial positions for learning and planning, and provides a quantitative framework that can stimulate further research.

# Materials and methods

## Data analysis

We re-analyze data from two publicly available data sets (*Mizuseki et al., 2013*; *Grosmark et al., 2016*) containing recordings of male Long-Evans rats running on linear tracks of various lengths for water rewards at both ends. From the first data set, we limited the analysis to the data from three animals, described in *Mizuseki et al., 2009*, which contained sufficient parallel recordings for Bayesian decoding of position. The second data set, described in *Grosmark and Buzsáki, 2016*, contains data from four animals running on a novel track. Data from one animal was excluded due to issues with local field potential recordings (see below), resulting in a total of six animals included in this study. The first 5 min of the recordings in a novel track were discarded since place fields of hippocampal place cells might be unstable in the first 5 min of novel experience (*Frank et al., 2004*) and we wanted to only include periods with stable spatial representations. Five of the animals had one to three experimental sessions available. The remaining animal had a disproportionately large number of experimental sessions, so a random subset was chosen. The exact number and identity of the sessions analyzed for each animal are reported in *Figure 6—figure supplement 2A*. In these data, neural activity was recorded using multi-shank silicon probes targeting region CA1 of dorsal hippocampus. After spike sorting, cells were classified as principal neurons or interneurons based on monosynaptic interactions, wave-shapes and burstiness.

### Local field potential

We filtered the local field potential (LFP) with a Butterworth filter of 3rd order between 4 and 12 Hz. We then estimated theta phase from the signal's Hilbert transform and shifted it by 180° so that 0° and 360° correspond to the peaks of the filtered LFP. Individual theta cycles were then defined as spanning the range from 0° to 360°. Periods of significant theta oscillation were identified by an instantaneous amplitude above the 97th percentile of amplitudes from a shuffled surrogate signal. To create this surrogate, we high-pass filtered the LFP at 1 Hz with another Butterworth filter of 3rd order and randomly permuted the time stamps. All the analyses carried out throughout the paper deal exclusively with periods with significant theta oscillations. In the excluded rat, the LFP signals were particularly noisy, leading to an atypically low fraction of theta cycles categorized as periods of significant theta oscillation as defined below ($0.38 \pm 0.02$, compared to an average of $0.81 \pm 0.06$ for the other animals).

### Tracking data

The position of the rat was determined as the midpoint of the two LEDs mounted on its head. The 2D positions where then projected onto the linear track. We calculated instantaneous speed as the central difference of the position along the track, smoothed with a Gaussian filter with a standard deviation of 100 ms. Periods where rats ran continuously in the same direction from corner to corner were identified as rightward or leftward runs. A characteristic speed was calculated for each running direction in 4 cm spatial bins by taking the mean of speeds after discarding instantaneous speed values below 10 cm/s that did not occur within 40 cm of either end of the track (e.g. see *Figure 6B*). This was done to avoid contamination of the calculation of the mean speed by infrequent cases where rats stopped in the middle of the track. Near the ends of the tracks, however, low speeds are the norm and therefore we did not set a minimum threshold there. A mean acceleration value was also calculated for each spatial bin by taking the difference of the smoothed speeds.

## Firing rates

After excluding spikes during pauses in track running (previous paragraph), firing rates were calculated in 4 cm spatial bins and smoothed with a Gaussian filter with a standard deviation of 6 cm. Virtually the same results were produced without this restriction (not shown). Firing rates were calculated independently for each running direction.

## Identification of place fields

Candidate place fields were required to have a peak firing rate above 2 Hz and at least 25 spikes. The spatial extent of the field was defined by the positions at which the firing rate fell below 15% of the peak firing rate. If a place field was cut on either side by the end of the track, but the firing rate had gone below 66% of the peak firing rate before then, the field was accepted but marked as incomplete. If there were more than three consecutive spatial bins with zero occupancy between the field's peak and one of the points at which the firing rate fell below threshold, the field was also marked as incomplete. Incomplete fields were excluded from the phase precession and place field skew analyses, and were treated differently in the place field size analysis (see below). Candidate place fields were screened manually to remove instances with multiple overlapping phase precession clouds.

## Place field sizes

For incomplete fields, place field size was calculated as twice the distance between the field's peak and the point on either side of the peak for which the firing rate fell below the 15% threshold. This ensured that a sufficient number of place fields at the ends of the track entered the analysis. Due to the pattern of place field skews along linear tracks (*Diba and Buzsáki, 2008*), we expect this method to slightly overestimate actual sizes, and thus the smaller place field sizes that we observed at the ends of the track should not be an artifact of the procedure.

For calculating the effect of running speed on place field sizes, we recalculated the firing rates with the same spatial bin size and smoothing as described above, but using data corresponding to periods with running speeds belonging to overlapping 20 cm/s wide speed bins in 10 cm/s steps starting at 2 cm/s. Rats spent too little time at some positions and speeds, often leading to an apparent firing rate of 0 due to the lack of time to produce any spikes. Thus, we defined a sampling index that identified areas within the spatial extent of the place field (defined above) that had been sufficiently sampled; we only attempt to measure the place field size in such cases. To obtain this sampling index, we summed the distances between all pairs of spatial bins within the field with occupancy above 300 ms, and divided it by the sum of distances between all pairs of bins in the field. This gave a number that increases nonlinearly from 0 (no valid bins) to 1 (all bins are valid) taking into account both the number of valid bins and how spread out they are through the field. We excluded place fields with a sampling index below 0.4. Place fields where then identified and their sizes calculated as described above. For the summary analyses, we removed extreme outliers below $Q1 - k \cdot IQR$ or above $Q3 + k \cdot IQR$ where Q1 and Q3 are the 25% and 75% percentiles, $IQR = Q3 - Q1$ and $k = 3$.

## Phase precession slopes

Phase precession slopes were measured from linear fits of phase precession clouds falling within the extent of place fields as defined above. The fitting of phase precession clouds is complicated by two matters: (i) phase is a circular variable, and (ii) the slope of the precession might be very steep. One possible way of dealing with the circularity of phase is to directly minimize angular distances between points and the best fitting line in the cylinder defined by position and phase. This can be done using circular-linear regression (*Kempter et al., 2012*; *Schmidt et al., 2009*) or by exhaustive search (*O'Keefe and Recce, 1993*). However, in such cases, one must set the minimum and maximum slopes allowed, otherwise the optimal solution would be a line with a slope approaching $\pm\infty$ that loops around the cylinder going through all points. This would preclude us from correctly fitting steep phase-precession clouds. Another possibility is to try to unwrap the phase precession cloud so that it lays uninterrupted in a plane and then perform a standard linear regression (*Lenck-Santini and Holmes, 2008*). However, this does not work well either for very steep phase precession

clouds, since linear regression minimizes only the sum of square errors in phase, leading to excessively shallow fits of steep clouds.

We therefore developed a new method to determine phase precession slopes robustly even when they are very steep. It is based on orthogonal distance regression, which minimizes the orthogonal distance of the data points to the best fitting line. This method introduces fewer a priori assumptions than standard linear regression, since it does not distinguish between independent and dependent variables. Phase and positions were normalized to a range of $[0, 1]$ to make their contribution equivalent in the calculation of orthogonal errors.

As commonly observed, spikes in the phase precession plot (theta phase vs. position) did not generally form a monotonic relationship. Instead, phase precession clouds slightly wrapped around 0° or 360°. This means that the theta phase, as extracted from the filtered LFP, does not accurately mark the transitions between actual theta cycles. Since we later analyze theta sequences, it is crucial to identify the proper beginning and end of the theta cycles. We did this by identifying the theta phase offset that optimized the phase precession relationship across all fields for each experimental session as follows. We considered potential offsets from −60° to 60° in 2° steps. For each value, we calculated orthogonal fits to the phase precession clouds for all fields within that session, and then selected the phase shift that led to the lowest mean orthogonal error across all fields. The optimal shift corresponded to the phase shift for which the most points fell along uninterrupted phase precession clouds.

In addition to ambiguity about boundaries of the theta cycle, for spikes near this boundary there is an ambiguity as to which cycle these spikes should be attributed to. This could lead to large errors in the orthogonal fit and therefore distortion in the estimate of the precession slope. Thus, we slightly modified the orthogonal fit procedure. We computed the slope and intercept of the line that minimized the sum of squared orthogonal distances with the following caveat. For points with normalized phases below 0.3 or above 0.7, we minimized the orthogonal distance to the line for either the original point or the point phase shifted in phase by 1 or −1, respectively, and chose whichever phase yielded the smallest distance. The minimization was done using the Nelder-Mead algorithm (*Gao and Han, 2012*) implemented in the SciPy library (*SciPy 1.0 Contributors et al., 2020*).

For calculating the effect of running speed on phase precession we proceeded as described for the place field sizes. We only attempted to calculate phase precession slopes when there were 12 or more spikes. Additionally, we analyzed phase precession slopes in single passes through a field. We calculated the average speed of the pass through the field from the tracking data. Note that this 'traversal speed' typically deviates from the 'characteristic speed' which is determined by the average over all the field traversals in a session. We focused on passes with a running speed above 2 cm/s, six or more spikes, a duration between the first and last emitted spikes above 400 ms and a coefficient of variation (standard deviation / mean) of instantaneous speed below 0.3. Passes with too many invalid positions in the tracking data where also excluded (sampling index < 0.4). For the summary analyses, we removed extreme outliers as described for place field sizes with $k = 3$ for phase precession slopes in the pooled phase precession analysis and $k = 6$ for single-pass phase precession slopes.

## Bayesian decoding and theta trajectory lengths

We performed a Bayesian decoding of position based on the ensemble spikes (*Zhang et al., 1998*) assuming a prior distribution over positions (*Davidson et al., 2009*). The probability of the rat being at position $x$ given a vector of spike counts for each cell, $n$, during a temporal bin of width $\tau$ is given by:

$$P(x|n) = C(\tau, n) \left( \prod_{i=1}^{N} f_i(x)^{n_i} \right) \exp\left( -\tau \sum_{i=1}^{N} f_i(x) \right) \tag{5}$$

where $C(\tau, n)$ is a normalization factor which can be determined by the normalization condition $\sum_x P(x|n) = 1$, and $f_i$ is the firing rate of cell $i$ in the corresponding running direction. We shifted theta phases so as to minimize the wrapping of phase precession clouds (see above), therefore we expected theta trajectories to be centered within individual theta cycles. We decoded within each cycle in overlapping 90° windows with a step size of 30° (no windows crossed theta cycles). The relatively large window size was due to the need to gather a sufficient number of spikes for the

decoding. These phase intervals were transformed into time intervals independently for each theta cycle based on the cycle's duration. We attempted to decode position only in time bins where at least one spike was emitted, and for positions bins within a spatial extent of 70 cm centered at the current position. This value is below the average place field size at all running speeds (*Figure 4C*) and should therefore encompass the whole extent of the sweep.

When averaging across theta cycles with similar speeds, we calculated theta trajectory lengths only in samples with more than five cycles. When analyzing theta trajectories in individual cycles, we focused on cycles well covered by phase bins with peak decoded probabilities larger than 0.1. In particular, we required at least five such phase bins spanning a minimum of 210˚. Theta trajectory lengths were derived from a best fit line. To calculate this best fit, we first found the line that maximized the sum of decoded probabilities for positions within ±5 cm of the line (*Davidson et al., 2009*) and then performed linear regression with the points in this region weighted by their probabilities.

### Place field skew

Place field skew was calculated as the ratio of the third moment of the place field firing rate distribution divided by the cube of the standard deviation (*Mehta et al., 2000*; *Davoudi and Foster, 2019*):

$$\widetilde{\mu}_3 = \frac{\mu_3}{\sigma^3}$$

$$\mu_3 = \frac{\sum_{j=1}^{S} f_j (x_j - \mu_1)^3}{\sum_{j=1}^{S} f_j}$$

$$\sigma = \sqrt{\frac{\sum_{j=1}^{S} f_j (x_j - \mu_1)^2}{\sum_{j=1}^{S} f_i}}$$

(6)

where, unlike above, $f_j$ is the firing rate of a given cell in the spatial bin $x_j$, $S$ is the number of spatial bins, and $\mu_1$ is the center of mass of the field:

$$\mu_1 = \frac{\sum_{j=1}^{S} f_j x_j}{\sum_{j=1}^{S} f_j}$$

(7)

## Mathematical description of coding schemes

### Spatial sweep

In the spatial sweep model, the represented position at time $t$, $r(t)$ is given by

$$r(t) = x(t) + d_\theta \frac{t - t_n}{T}$$

(8)

where $x(t)$ indicates the actual location of the animal, $d_\theta$ represents the extent of the spatial sweep, $T$ is the period of the theta oscillation and $t_n$ is the time for which $r(t) = x(t)$ within the current theta cycle.

In order to derive an expression for the length of theta trajectories, we can perform a Taylor series expansion around $t_n$:

$$r(t) = x(t_n) + v(t_n)(t - t_n) + \frac{a(t_n)}{2}(t - t_n)^2 + ... + d_\theta \frac{t - t_n}{T}$$

(9)

where $v$ and $a$ are the speed and acceleration, respectively. Since $(t - t_n)$ is small, we can ignore the effects of the acceleration and higher order derivatives and approximate the represented position as:

$$r(t) = x(t_n) + v(t_n)(t - t_n) + d_\theta \frac{t - t_n}{T}$$

(10)

Here, the first two terms reflect the actual movement of the animal and the last one represents the sweep across positions starting behind the current position of the animal and extending ahead. Note that the model makes the simplifying assumption that sweeps, and hence also phase precession, are linear. There are indications that this could be the case when analyzing single trials

(*Schmidt et al., 2009*) or when calculating theta phase using a wave-form method (*Belluscio et al., 2012*) (but see *Souza and Tort, 2017*; *Yamaguchi et al., 2002*).

From *Equation 10,* we can calculate the length of theta trajectories, which encompasses both the sweep and the change in the actual position of the animal due to its movement during a theta cycle:

$$l = r(t_n + (1 - \alpha)T) - r(t_n - \alpha T) = v(t_n)T + d_\theta, \tag{11}$$

where $\alpha$ is the relative position within the theta cycle for which $r(t) = x(t)$. The length of theta trajectories has an offset of $d_\theta$ and is only weakly dependent on speed since the speed term is multiplied by the relatively small theta period.

At the single-cell level, we postulate a cell's activity is driven by the represented position $r(t)$ according to its true place field with preferred location $r_0$ and width $\sigma$. Specifically,

$$a(r) = f_{\max} \exp\left(-\frac{r - r_0}{2\sigma^2}\right). \tag{12}$$

All theta sweep models postulate that the position $r(t)$ represented by the place cell population differs from the actual location of the animal $x(t)$ depending on the phase of theta. $r(t) - x(t) = -\alpha d_\theta$ at the beginning of the theta cycle, and $r(t) - x(t) = (1 - \alpha)d_\theta$ at the end. Since a place cell's place field is determined by $r(t)$ at all times, its firing relative to $x(t)$ will shift with the theta phase and this gives rise to phase precession in this model. For instance, the peaks of firing occur at $r_0 - \alpha d_\theta$ at the beginning of the theta cycle and at $r_0 + (1 - \alpha)d_\theta$ at the end of the cycle. Hence, the theta phase of maximal firing precesses over a range of $d_\theta$ and the slope of phase precession is:

$$m = -\frac{360°}{d_\theta} \tag{13}$$

Furthermore, the shift of the cell's firing enlarges the measured place field beyond its true place field (size $s_0$) by the amount of the shift, that is the size of the measured place field is given by

$$s = s_0 + d_\theta \tag{14}$$

However, in practice, this will be further modulated by other factors such as the presence of theta modulation or the smoothing used when calculating firing rates.

## Temporal sweep

For the temporal sweep model, the represented position is given by,

$$r(t) = x\left(t + \tau_\theta \frac{t - t_n}{T}\right) \tag{15}$$

where $\tau_\theta$ is the extent of the temporal sweep. Again, we can perform a Taylor series expansion around $t_n$:

$$r(t) = x(t_n) + v(t_n)\frac{(\tau_\theta + T)}{T}(t - t_n) + \frac{a(t_n)}{2}\frac{(\tau_\theta + T)^2}{T^2}(t - t_n)^2 + \dots \tag{16}$$

And discarding the effects of higher-order derivatives:

$$r(t) = x(t_n) + v(t_n)(t - t_n) + v(t_n)\tau_\theta\frac{t - t_n}{T} \tag{17}$$

where again the first two terms reflect the movement of the animal and the last one, the sweep. In this case, the length of the theta trajectories is directly proportional to running speed:

$$l = r(t_n + (1 - \alpha)T) - r(t_n - \alpha T) = (\tau_\theta + T)v(t_n) \propto v(t_n) \tag{18}$$

From the first-order approximation of $r(t)$ in *Equation 17,* we see that the spatial extent of the sweep is $v(t_n)\tau_\theta$. Therefore, we can substitute this for $d_\theta$ in *Equation 13* and *Equation 14* to obtain the equations for phase precession slopes and place field sizes:

$$m = -\frac{360°}{v(t_n)\,\tau_\theta} \propto -\frac{1}{v(t_n)}, \tag{19}$$

and

$$s = s_0 + v(t_n)\,\tau_\theta \tag{20}$$

As speed goes to 0, phase precession slopes becomes increasingly steep and place field sizes converge to that of the underlying true place fields.

## Behavior-dependent sweep

To reconcile the presented analytical observations with aspects of spatial and temporal sweeps that they appear to be consistent with, we make the extent of spatial sweeps directly proportional to the characteristic running speed at each location. Thus, we set $d_\theta = \overline{v}(x(t))\,\tau_\theta$ in *Equation 8*:

$$r(t) = x(t) + \overline{v}(x(t))\,\tau_\theta\,\frac{t - t_n}{T}. \tag{21}$$

Note that second term representing the sweep is equivalent to the sweep term in *Equation 17*, but using the characteristic speed through the current location as opposed to the instantaneous speed at which the animal is running.

Proceeding as above, we can derive equations for the theta trajectory length:

$$l = v(t_n)T + \overline{v}(x(t_n))\,\tau_\theta, \tag{22}$$

which is roughly proportional to the characteristic running speed. Similarly, we can obtain the equations for the phase precession slopes, $m$:

$$m = -\frac{360°}{\overline{v}(x(t_n))\,\tau_\theta}, \tag{23}$$

and place field sizes:

$$s = s_0 + \overline{v}(x(t_n))\,\tau_\theta. \tag{24}$$

Place fields will therefore be larger and have shallower phase precession slopes for locations with higher characteristic running speed, but will not otherwise vary with instantaneous running speed.

## Computational modeling

We generated surrogate data sets by taking the tracking and LFP data from experimental sessions and substituting the experimental spikes with those generated by our models under different coding assumptions. We discretized the session into 2 ms time bins and calculated for each bin the position that the cell population represents, $r(t)$, according to each coding scheme at that particular phase of theta, $\theta(t)$. For the temporal sweep model, $r(t) = x(t + \tau_\theta(\theta(t) - 180)/360)$, where $x(t)$ is the position obtained from the tracking and $\tau_\theta$ is the time interval swept over in each theta cycle. We manually tuned $\tau_\theta$ iteratively to minimize the deviations in theta trajectory lengths, place field sizes and phase precession slopes between experimental and simulated data (*Figure 7*). A good compromise was obtained for a $\tau_\theta$ of 0.55 s. For the spatial sweep model, $r(t) = x(t) + d_\theta(\theta(t) - 180)/360$, where $d_\theta$ is the spatial distance swept over in each theta cycle. We tuned $d_\theta$ as described for $\tau_\theta$, obtaining a value of 30 cm. For the behavior-dependent sweep model, $r(t) = x(t) + \overline{v}(x(t))\,\tau_\theta(\theta(t) - 180)/360$, where $\overline{v}(x(t))$ was taken to be the experimentally observed characteristic running speed at each position, calculated independently for each experimental session and running direction (e.g. *Figure 6A*). $\tau_\theta$ was obtained from fitting *Equation 23* to the relationship between inverse phase precession slopes and characteristic running speed (*Figure 6I*), resulting in a value of 0.57 s.

Then, for each time bin, we generated spikes for each of 20 simulated place cells uniformly distributed along the track. Spikes were generated based on the activation level of each cell $i$ at the represented position, as determined by the cell's true place field. These true place fields were defined as Gaussian activation functions, $f_i$, with a peak value of 1, discretized in 1 cm steps. The standard deviation of the Gaussian, $\sigma$, determines the size of the true place fields, which controls

how tight the relationship is between firing at a particular phase and the time or distance to the peak's center. A small true place field size leads to very crisp phase precession clouds, whereas a bigger one results in wider clouds with a noisier relationship between phase and position. For the temporal and spatial sweep models, the standard deviation was set to 7 cm, which corresponds to true place field sizes of 27 cm. With this value, the model produced phase precession clouds approximately matching the experimental ones. For behavior-dependent sweeps, the standard deviation was heterogeneous across the population, set to $\max(4, 0.3\overline{v}(c_i)\tau_\theta)$, where $c_i$ is the center of each place field.

Spikes were generated from a Poisson process with probability equal to an instantaneous firing rate times the width of the time bin:

$$P_i(t) = [15 + 0.2v(t)][1 - m_\theta \cos(\theta(t))]f_i(r(t))\delta t, \tag{25}$$

where $v(t)$ is the instantaneous running speed in cm/s. The first term sets the maximum firing rate and the values were chosen to approximate the increase in peak firing rate with running speed observed in the experimental data (*Figure 7—figure supplement 2*). The second term introduces an inhibitory modulation by the amplitude of the theta oscillation with $m_\theta = 0.35$ approximating the experimentally observed value in the experimental data set (not shown).

For the model predictions shown in the the schematics in *Figure 1* and *Figure 2*, we simulated runs at constant speed with a regular 8 Hz theta oscillation, and used the models as described above, except for $\tau_\theta = 0.5$ s and $\sigma = 6$ cm.

## Acknowledgements

We thank Loren Frank for helpful discussions and Laia Serratosa Capdevila for feedback on the draft. We thank the anonymous reviewers for their thoughtful and extensive feedback which led to a clearer and more comprehensive article.

## Additional information

### Funding

| Funder | Grant reference number | Author |
|---|---|---|
| German Research Foundation | 419037518 - FOR 2812,P2 | Sen Cheng |
| Federal Ministry of Education and Research | 01GQ1506 | Sen Cheng |
| National Institute of Mental Health | NIMH R01MH109170 | Kamran Diba |

The funders had no role in study design, data collection and interpretation, or the decision to submit the work for publication.

### Author contributions

Eloy Parra-Barrero, Conceptualization, Software, Formal analysis, Visualization, Methodology, Writing - original draft, Writing - review and editing; Kamran Diba, Conceptualization, Supervision, Funding acquisition, Methodology, Writing - review and editing; Sen Cheng, Conceptualization, Formal analysis, Supervision, Funding acquisition, Methodology, Writing - original draft, Project administration, Writing - review and editing

### Author ORCIDs

Eloy Parra-Barrero https://orcid.org/0000-0001-5702-4466
Kamran Diba https://orcid.org/0000-0001-5128-4478
Sen Cheng https://orcid.org/0000-0002-6719-8029

### Decision letter and Author response

Decision letter https://doi.org/10.7554/eLife.70296.sa1

Author response https://doi.org/10.7554/eLife.70296.sa2

# Additional files

## Supplementary files

- Supplementary file 1. Summary of statistical tests.
- Transparent reporting form

## Data availability

No new data was generated for this study. All analysis code has been made available at https://github.com/sencheng/models-and-analysis-of-theta-phase-coding (copy archive at https://archive.softwareheritage.org/swh:1:rev:cadbae0ff9a9f1af82101229c15f4af213857773).

The following previously published datasets were used:

| Author(s) | Year | Dataset title | Dataset URL | Database and Identifier |
|---|---|---|---|---|
| Mizuseki K, Sirota A, Pastalkova E, Diba K, Buzsáki G | 2013 | Multiple single unit recordings from different rat hippocampal and entorhinal regions while the animals were performing multiple behavioral tasks | http://dx.doi.org/10.6080/K09G5JRZ | Collaborative Research in Computational Neuroscience, 10.6080/K09G5JRZ |
| Grosmark AD, Long J, Buzsáki G | 2016 | Recordings from hippocampal area CA1, PRE, during and POST novel spatial learning | http://dx.doi.org/10.6080/K0862DC5 | Collaborative Research in Computational Neuroscience, 10.6080/K0862DC5 |

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

## Appendix 1

### Variable theta phase locking fails to account for the combination of population and single-cell results

For completeness, we also considered whether the experimental results could be accounted for by a fourth model similar to the one proposed by *Chadwick et al., 2015*. Cells participate in spatial sweeps, but the coordination between cells increases with running speed. At lower speeds, the lack of coordination between cells should bias the decoder toward the current position, leading to shorter decoded theta trajectories.

We modeled this by introducing the following modifications to the spatial sweep model. To increase the coordination between pairs of cells at the theta timescale with speed, we introduced into each cell a certain amount of independent theta phase noise that decreased with running speed:

$$\theta_i(t) = (\theta(t) + N(0, \sigma_\theta)) \mod 360$$
$$\sigma_\theta = ke^{-\gamma v(t)},$$

(26)

where $N(0, \sigma_\theta)$ are samples obtained from a Gaussian distribution with standard deviation $\sigma_\theta$; with $k = 120$ and $\gamma = 0.025$. This is the crucial feature of the model. Additionally, to get a more realistic amount of variance in place field sizes and phase precession slopes, we used a slightly different theta distance for each cell:

$$d_{\theta,i} = \max\{0, \sim d\alpha_i + \beta_i\},$$

(27)

with $d = 35$, and $\alpha_i$ and $\beta_i$ derived from Gaussian distributions $N(1, 0.3)$ and $N(0, 10)$, respectively, and fixed for each cell for the entire simulation. The standard deviation setting the width of each true place fields was $\max(3, 0.25 d_{\theta,i})$.

As expected, the model provides a good fit for the increase in theta trajectory lengths with running speed. However, the model does not account for the increase in average place field sizes (*Appendix 1—figure 1A,B*). Furthermore, unlike the experimental data, the model produces atypically noisy phase precession clouds at low running speeds (*Appendix 1—figure 1C*) which become sharper with speed, as measured by decreasing residuals in the phase precession cloud fits (*Appendix 1—figure 2A,B*).

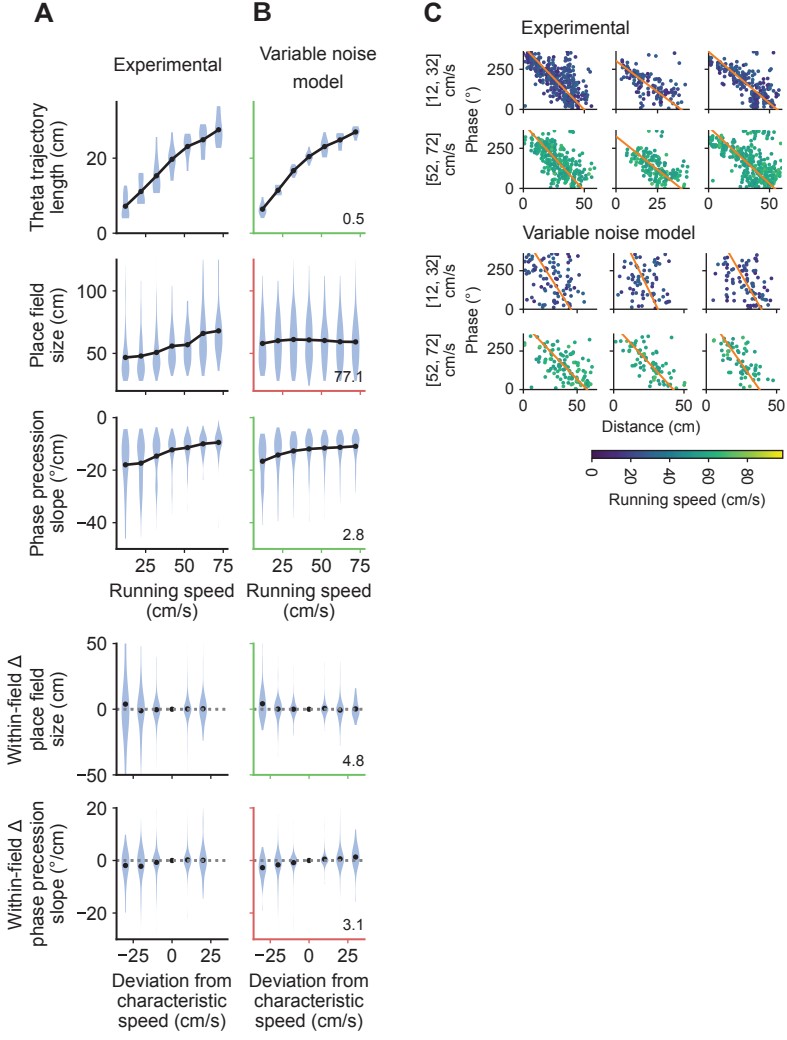

**Appendix 1—figure 1.** Variable theta phase locking fails to account for the combination of population and single-cell results. (**A**, **B**) Comparison of experimental and simulated data produced by a model with constant $d_\theta$ but variable theta phase noise. Plotting conventions as in *Figure 7*. The variable noise model does a good job at capturing the increase in theta trajectory lengths and phase precession slopes with running speed, but it does so at the cost of modest within-field changes in phase precession slopes and without capturing any of the increase in place field sizes. (**C**) Example phase precession clouds at low and high speeds for experimental and modeled place fields. Each column corresponds to a cell. The variable noise model introduces an atypically large amount of noise at low speeds as compared to the experimental data.

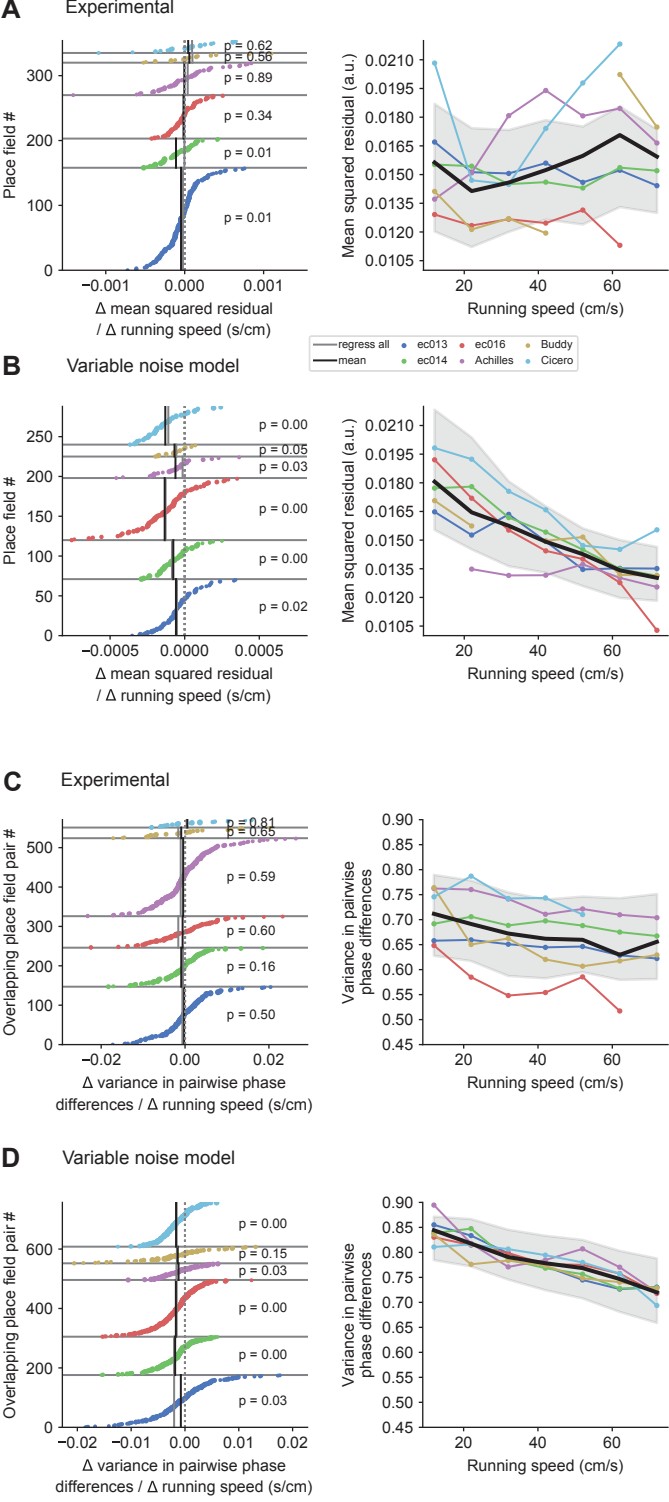

**Appendix 1—figure 2.** Theta phase locking and theta timescale coordination between cells vary with speed in the variable noise model but not in the experimental data. (**A**, **B**) The effect of speed on the residuals of the phase precession cloud fits (i.e. mean squared orthogonal distances from the points to the fitting line). The fits are calculated after normalizing the place field size and the phase precession range (360°) to 1, so the residuals are dimensionless. In the variable noise model, the

*Appendix 1—figure 2 continued on next page*

*Appendix 1—figure 2 continued*

residuals decrease systematically with speed, as the phase vs. position relationship of spikes becomes sharper. This relationship, however, is not clear in the experimental data. Plotting conventions as for *Figure 4B,C*. (**C, D**) The effect of speed on the circular variance in phase differences between spikes emitted by pairs of cells with overlapping place fields. The variable noise model produces atypically high variance, and the variance tends to decrease with speed for individual cell pairs, which is not the case for the experimental data.

Even if phase precession clouds were noisier at lower speeds, cell coordination, and hence theta trajectory lengths could remain unaffected, if noise affected all cells in the same way. Therefore, we measured the effect of speed on cell coordination directly, by calculating the variance in phase differences between spikes of pairs of cells with overlapping place fields. First, we found cells whose place fields overlapped by more than 20 cm. Then, when rats were within 10 cm of the overlapping region of two cells, we collected, for each theta cycle individually, the phases at which spikes from each cell were emitted, and calculated all pairwise phase differences between them. Finally we calculated their circular variance (*Fisher, 1993*) as:

$$V = 1 - \overline{R} = 1 - \sqrt{\left(\frac{1}{N}\sum_{i=1}^{N}\cos\phi_i\right)^2 + \left(\frac{1}{N}\sum_{i=1}^{N}\sin\phi_i\right)^2},\tag{28}$$

where $\phi_i$ is $i$-th pairwise phase difference between the two cells.

If the cells code for positions in a coordinated fashion, the variance in phase differences between their spikes should be low. We found that the level of variance present in the variable noise model at low speeds was unrealistically high as compared to the experimental data (*Appendix 1—figure 2C,D*). Furthermore, in the experimental data, the variance in phase differences for spikes of a cell pair did not significantly change with running speed for any of the animals ($p>0.15$; Wilcoxon signed-rank test), whereas it did so for five of the six animals in the simulated data ($p<0.05$; Wilcoxon signed-rank test).

Overall, we conclude that these results rule out the possibility that changes in cell coordination at the theta time scale play a major role in explaining the changes observed in theta trajectory lengths.

## Appendix 2

### Summary of statistical tests

*Supplementary file 1* contains a table with a summary of all statistical tests performed for the analyses of *Figure 4*, *Figure 5* and *Figure 6*. The table indicates animal names, number of samples (N), test types and p-values.

