## [Decision Letter]

**Acceptance summary:**

This work develops a novel and rigorous mathematical framework to analyze spatial and temporal sequence coding in hippocampal place cells. The findings have potential to advance our understanding of hippocampal place cell functions by providing an explanation for seemingly conflicting results in previous studies of theta phase precession and theta sequences. The findings are likely to have an important impact on the hippocampal community.

**Decision letter after peer review:**

Thank you for submitting your article "Neuronal sequences during theta rely on behavior-dependent spatial maps" for consideration by *eLife*. Your article has been reviewed by 3 peer reviewers, and the evaluation has been overseen by a Reviewing Editor and Laura Colgin as the Senior Editor. The reviewers have opted to remain anonymous.

The reviewers have discussed their reviews with one another, and the Reviewing Editor has drafted this to help you prepare a revised submission. A list of Essential Revisions, compiled from all reviews, is provided below. Please refer to the individual reviews, shown in their entirety below, for additional details about the comments in the essential revisions.

Essential revisions:

1. The authors conclude the experimental data best fit the behavior-dependent sweep model. However, the predicted effects on theta trajectory length and place cell properties are very similar between the temporal sweep model and the behavior-dependent sweep model (Figure 7). Therefore, it is unclear if they can conclusively determine experimental data best fit the behavior-dependent sweep model rather than the temporal sweep model, especially based on visual inspection. Their argument would be strengthened by analytically directly comparing the models and experimental data to determine which is a best fit. For example, is there are larger different between predicted and real data for the spatial and temporal sweep models than the behavior-dependent sweep model? A deeper analysis of what specifically differentiates the temporal sweep and behavior-dependent sweep is warranted. For example, the authors state "The temporal sweep model could be fitted to roughly capture the observed increase in theta trajectory lengths, place field sizes and phase precession slopes pooled by running speed, but it incorrectly predicted changes in individual fields with speed." Visually these "incorrect predictions" do not seem that far off. Are they significantly different and if so what is the magnitude of the difference?

2. The behavior-dependent sweep model (equation 3) is essentially the spatial sweep model with dependence on the averaged speed which is a function of location. There is no instantaneous speed term in this model, even though they show the effects of instantaneous running speed in the model (Figure 2). The authors do not describe how exactly variation in instantaneous speed affect the model. Does the model only use average speed and if so, over what time scales? If the model uses average speed over long timescales like a whole trial or multiple trials, how does the model account for rapid changes in speed levels, like at the scale of a single theta cycle? Relatedly, the use of the speed term needs to be clarified in several places:

a. In Figure 7D, cell firing properties are separated for different speed levels. Over what time scales was speed measured? Are these speed levels the "average speed" term in equation (3)?

b. Based on equation (21), phase precession slope depends on the average speed in the behavior-dependent sweep model. How does this fit on single trial phase precession slope? For single phase precession, is the "averaged speed" calculated from the limited data within that trial?

3. Related to the above points, the two models that the authors contrast with their own with are oversimplified. Rather than highlighting the virtues of their own model, this creates the impression that any model in which place cells do not have unrealistic properties, such as place field width vanishing as the animal slows down, would work equally well. This impression is reinforced by the fact that crucial aspects of the model are not tested, and instead very general properties, some of which were already known, are tested as predictions of the model: running speed and properties of place cells vary along the track. To increase the impact of this work, the text should be revised to ensure that the authors' model is not compared with overly simplified models.

4. The authors should make sure that prior work from other labs is presented in its proper context and not specifically associated with one of the two overly simplified models presented in the text.

5. To address the issue whether field size or speed is better correlated with theta (spatial) length, the authors may consider producing a plot similar to Figure 7A, but using (mean) place field size rather than (mean) running speed, and adding a simulation condition using mean place field size as a controlling variable. The outcome could be one is better than the other, or they are similar or equivalent. Regardless of the outcome, the authors could discuss further the functional differences between these scenarios.

6. What is the evidence for the assumption of a "true place field"? What are the potential limitations of this assumption?

7. To increase impact, the overall organization and presentation of ideas should be revised to ensure clarity for a general audience.

*Reviewer #1:*

In this work, the authors develop precise mathematical models to determine if sequential hippocampal neural activity during behavior is best described as spatial or temporal sweeps. While prior work has shown evidence relating neural activity to both spatial and temporal sweeps, a cohesive and quantitative description of how neural activity sweeps relate to both time and space is needed. Whether theta sweeps better represent spatial or temporal coding of position (eg distance vs time to past and future locations) is readily distinguishable when animals run at different speeds. The authors make use of these models to make specific predictions about how population codes and single cell properties will vary with animal speed. The authors find that neither the spatial sweep model nor temporal sweep model fully describes observed place cells firing properties at the single cell or population levels. They propose a 'behavior-dependent' sweeps model to bridge both spatial temporal models and claim this model best described the experimental observations. Overall, this work develops a novel and rigorous mathematical framework to study spatial and temporal sequence coding in hippocampus and uses this framework to test whether theta sweeps in the hippocampus best represent spatial or temporal codes. Therefore, this work addresses important questions about how the hippocampus simultaneously encodes spatial and temporal information.

However, some clarifications are required. Specifically, it is unclear if they can conclusively determine experimental data best fit the behavior-dependent sweep model rather than the temporal sweep model, especially based on visual inspection. Their argument would be strengthened by analytically directly comparing the models and experimental data to determine which is a best fit. Furthermore, the temporal resolution of speed used in the models and analysis of experimental data requires clarification. In particular, it is unclear if the temporal resolution of speed is precise enough to account for rapid changes in speed.

1. The authors conclude the experimental data best fit the behavior-dependent sweep model. However, the predicted effects on theta trajectory length and place cell properties are very similar between the temporal sweep model and the behavior-dependent sweep model (Figure 7). Therefore, it is unclear if they can conclusively determine experimental data best fit the behavior-dependent sweep model rather than the temporal sweep model, especially based on visual inspection. Their argument would be strengthened by analytically directly comparing the models and experimental data to determine which is a best fit. For example, is there are larger different between predicted and real data for the spatial and temporal sweep models than the behavior-dependent sweep model? A deeper analysis of what specifically differentiates the temporal sweep and behavior-dependent sweep is warranted. For example, the authors state "The temporal sweep model could be fitted to roughly capture the observed increase in theta trajectory lengths, place field sizes and phase precession slopes pooled by running speed, but it incorrectly predicted changes in individual fields with speed." Visually these "incorrect predictions" do not seem that far off. Are they significantly different and if so what is the magnitude of the difference?

2. The behavior-dependent sweep model (equation 3) is essentially the spatial sweep model with dependence on the averaged speed which is a function of location. There is no instantaneous speed term in this model, even though they show the effects of instantaneous running speed in the model (Figure 2). The authors do not describe how exactly variation in instantaneous speed affect the model. Does the model only use average speed and if so, over what time scales? If the model uses average speed over long timescales like a whole trial or multiple trials, how does the model account for rapid changes in speed levels, like at the scale of a single theta cycle? Relatedly, the use of the speed term needs to be clarified in several places:

a. In Figure 7D, cell firing properties are separated for different speed levels. Over what time scales was speed measured? Are these speed levels the "average speed" term in equation (3)?

b. Based on equation (21), phase precession slope depends on the average speed in the behavior-dependent sweep model. How does this fit on single trial phase precession slope? For single phase precession, is the "averaged speed" calculated from the limited data within that trial?

3. What is the evidence for the assumption of a "true place field"? What are the potential limitations of this assumption?

*Reviewer #2:*

After the finding of theta phase precession in rat place cells, it was soon discovered that place cells form sequential firing patterns within individual theta cycles that maintain the sequential order of their place fields along a spatial trajectory. The so-called theta sequences have important functional consequences, because they indicate that place cells at the population level encode not just the animal's current location, but also the past and future locations, all amazingly in the short time scale of a theta cycle (~120 ms). In this paper, the authors ask whether the "past and future" coding is spatial or temporal, i.e., whether the spatial contents represented in theta sequences go beyond a fixed time or distance from the animal's current time or location. The authors realize that results from previous studies cannot be explained by either, especially those regarding how the spatial distance within a theta sequence depends on the animal's running speed. The authors present a new explanation, called "behavior-dependent sweep model", which relies on a key point that place fields of individual place cells vary in their sizes (lengths) at different locations along a trajectory and they tend to be larger when speed is high. By analyzing existing data and by computational modeling, the authors present evidence that supports their model.

Although most of the results from the data analysis confirm the findings in previous studies, these results, together with the key new finding on the relationship between place field size and speed (Figure 6), support the authors' conclusion. The data analysis and modeling are straightforward, the writing is mostly clear, and the conclusion is well balanced with sufficient discussions.

However, there is one major concern that may help clarify or strengthen the author's model that emphasizes the speed at each location of a trajectory, which the authors conceptualize as varies around a "characteristic" (mean) speed. The question here is whether it is necessary to introduce a behavioral variable, rather than a neurophysiological or functional one, to explain the spatial contents in theta sequences. A related and perhaps important issue is which one, place field size or speed, is more fundamental here. It is possible that simply stating that field sizes systematically vary at different locations along a trajectory and the animal tends to run faster at locations with larger fields, together with the well-known properties of of single-cell theta phase precession (termed "spatial sweep" by the authors), could be sufficient to explain all the data. The reason why emphasizing place field size is that place field is a functional concept that could be influenced by factors in spatial information encoding, such as task demand (e.g. turning points, reward sites), sensory cue density (e.g. rich in visual cues), or spatial layout (e.g. long straight arm). It is conceivable that these factors systematically vary from one location to another, leading to systematically larger or smaller place fields. On the other hand, running speed varies from one animal to another, from one session to another session in the same animal, and from one moment to another moment (and thus the definition of a "characteristic" mean speed is needed). From the data in Figures 3B, 6B, it seems that the same animal had more speed variation in one session (ec014, green), but the correlation between theta (spatial) length and speed was worse, than another (ec016, red). It is possible that the data like these can be better explained by directly targeting the relationship between theta length and place field size.

1) I only have one major suggestion. To address the issue whether field size or speed is better correlated with theta (spatial) length, the authors may consider producing a plot similar to Figure 7A, but using (mean) place field size rather than (mean) running speed, and adding a simulation condition using mean place field size as a controlling variable. The outcome could be one is better than the other, or they are similar or equivalent. Regardless of the outcome, the authors could discuss further the functional differences between these scenarios.

*Reviewer #3:*

Parra-Barrero and colleagues analyze and compare conceptually different modes in which neuronal sequences could be organized within a theta cycle in the hippocampus. They show that we cannot think of them as purely temporal or purely spatial predictions, and instead propose that we should think of them as temporal predictions at the animals' characteristic speed in a given position, hence 'behavior-dependent' sweeps. While this manuscript has many strengths and represents a significant step forward in organizing ideas around theta sweeps, my feeling is that it compares a reasonable model with two oversimplified models. More work should be directed to understanding which are the crucial ingredients for this model to work, and what are the degrees of freedom that could be used to build a family of models that would perform equally well.

Parra-Barrero and colleagues organize existing ideas about theta sweeps into a framework that allows contrasting different models with key experimental observations. In particular, they show that purely spatial theta sweeps (where the extent of the spatial sweep, and thus the place field size, is the same for all locations) do not produce a dependence of place field size or phase precession slope on running speed, as observed in experimental data. Similarly, they show that purely temporal sweeps (projecting the current instantaneous trajectory of the animal for a fixed time interval), produce sweeps with a length proportional to running speed. As running speed approaches zero, this generates the unrealistic situation in which theta sweep length and place field width go to zero, while phase precession slope goes to minus infinity. The authors contrast these two models with one based on behavior-dependent sweeps. In this model, sweeps are temporal but their spatial extension is proportional to the typical running speed in that position.

The major strengths of the manuscript are two. First, organizing previous ideas into a framework where different models can be contrasted. Second, proposing a model that does not have the caveats of the purely temporal or the purely spatial sweeps. I find that it also has some weaknesses. First, the two models that they contrast their own with are oversimplified. Rather than highlighting the virtues of their own model, this creates the impression that any model in which place cells do not have unrealistic properties, such as place field width vanishing as the animal slows down, would work equally well. This impression is reinforced by the fact that crucial aspects of the model are not tested, and instead very general properties, some of which were already known, are tested as predictions of the model: running speed and properties of place cells vary along the track. Second, I think that the authors have gone too far in associating the two oversimplified models with the previous work of colleagues that, in my view, not only would not accept these ideas as their own but in some cases would even claim that they have already published some results contradicting them. Carefully dissecting exactly which ideas were put forward by whom in the past would not only be fair to colleagues in the field but would also highlight the contrasting novelties presented in this manuscript. Third, the overall development of ideas is somewhat obscure and I think that the authors could find a way to present information so as to maximize the chances that interested readers will understand everything on the first attempt. The authors have chosen a way to present their ideas in which contradictions are left unresolved for a substantial part of the paper. In my view, this aesthetic choice has the risk of losing readers before they get to the point where everything becomes clear (the behavior-dependent sweep model and the idea that different cells provide information about different parts of the track, with different typical speed and place field properties). There is no need to do this, contradictions could be resolved as they appear, helping the reader to better navigate the manuscript.

The main result of the authors is a model that is compatible with experimental results. In my view, they should put some effort into clarifying whether this is a unique model or similar proposals would perform equally well. This means dissecting the crucial aspects that make this model better than any other (not just the two oversimplified models with unrealistic place field properties). Variations that I can think of are, for example, temporal sweeps with an overall mean speed rather than the mean speed for a given position, or spatial sweeps with a d_theta that depends on the width and density of place fields. If they find that there is a family of models sharing some crucial element that perform similarly, this would strengthen rather than weaken the manuscript. In addition, they should test specific predictions of the behavior-dependent sweep model. For example, characteristic speed and theta trajectory length along the track are not proportional (Figures 6B and C). Does the model require some improvement to account for this?

Overall I think that the authors have succeeded in bringing into the spotlight properties of theta sweeps that have been studied extensively from a qualitative perspective, but for which quantitative accounts are lacking. They have put forward a framework in which different models can be compared and proposed a model that is compatible with most of the experimental observations. In this sense, I think that a revised version of this manuscript would have an important impact on the hippocampal community.

30 – 'The current position of the animal'. There is an issue in how one defines this. What we typically have is an estimation of the position of LEDs that are placed somewhere above the animal's head. We ignore the distance between these LEDs and the position that CA1 considers to be the 'actual' position of the animal. Please comment on how this correction would affect the estimation of past and future positions at different speeds, especially when considering time sweeps.

69 – Geisler and colleagues (2007) claim in page 1:'Furthermore, the phase advance of the spikes relative to theta was steeper in fast than in slow trials'.

94 – Again, Geisler et al., 2007, seem to claim the opposite. On top of this, Maurer et al., 2012 do not seem to refer directly to the phase precession slope. They find a relationship between speed and look ahead distance and a speed dependent population CCG slope. Somehow this does not seem to comply with what is described here as the spatial sweep model not accounting for results at the population level. Perhaps what the authors present as 'the model' and the bits of it that can be related to previous descriptions could be described in greater depth.

98 – This length does not seem to be affected by a potential increase in field size as a result of place cells increasing their firing rate with speed. Why is this not taken into account?

625 – Equation 12. It should be clarified that this is an approximation, since phase precession has been shown to be highly non-linear already by O'Keefe and Recce. Thus, the exact value of the slope is highly dependent on the method, especially on the relative sampling of different phase ranges.

108 – L = v(t). As I understand it, dimensions do not match. The same happens to Equation 16 in the methods. Do parenthesis stand for multiplication or evaluation of the function v at a given time?

635 to 637 – There is a problem with the logic here. As v, goes to 0, v*tau going to 0 implies a slope going to minus infinity. However, v*tau is roughly speaking the width of the field of a cell at speed v. I do not think any of the authors cited in this manuscript would claim that the width of fields vanishes as v goes to 0. What is this temporal sweep model actually modeling? The fact that tau is independent of speed is more a choice of the manuscript authors than a claim coming from experimental results.

108 – 'This is almost exactly the result reported by Maurer et al., (2012); Gupta et al., (2012).'. Please specify how and in which figures. For example, Figure 8 from Gupta et al., seems to be at odds with this claim.

112 – Please specify or give an example of which procedures or methods can cause a contradiction. It is not clear from the text.

Figure 4a – The visualization of these examples is really poor. Since the purpose of this panel is visualization, I suggest to find a better way, perhaps with fewer examples or speed bins.

Figure 4b-c – Although the conclusion is plausible, and the examples seem to indicate this, the analysis is not convincing. The procedure based on regressing 7 datapoints has too much error, and this might mask an effect at the population level. (For example, are these individual regressions significant?) A high p-value does not necessarily mean that there is no effect. I suggest trying other methods. For example, the half-width decay in the spatial autocorrelation for the population of all cells could be studied and compared at different speeds. Fewer speed groups would also help getting better statistics. All of the analyses could also be applied to pools of cells with neighboring fields, to explain the contradiction right a way and thus help the reader navigate the manuscript.

178 – It is not clear at this point of the manuscript why both results are compatible with behavior dependent sweeps. This might need some more explanation.

191 – I suggest to try to make explicit and consistent the description of slopes. Unfortunately, describing a negative slope as 'lower' is ambiguous. Similarly, in 209 an 'increase in the slope' is mentioned, which is also ambiguous. Please revise this language throughout the manuscript. I believe people in the past have referred to slopes as shallower or steeper to avoid ambiguity.

181 – 194. What would the different models predict and why?

234 – Why are these results striking? Similar findings have been reported before. For example in Gupta et al., 2012. I disagree with lines 312 – 314. Figure S3 of Gupta et al., shows very similar results to the ones analyzed here.

Figure 6A – If this explains the apparent contradictions found in Figures 4 and 5, then pooling locations with similar average speed should make the pooled average effects disappear. This needs to be shown in those figures to help readers navigate the manuscript.

Figure 6,B-C – running speed at 0.25 is leeds then two thirds of the speed at 0.75. However theta length trajectories are similar. To my view this speaks directly against the simple explanation that is being tested 'Short theta trajectories and small place field sizes could occur where animals tend to run slow, contributing to the low speed averages, and long theta trajectories and large place field sizes could occur where animals tend to run fast, contributing to the fast speed averages'. There seems to be something else going on here.

Figure 7 – Comparisons are purely visual. Could some quantification be provided?

329 – Figure S5. In this paragraph an alternative model is discarded, but no quantification is provided. In the last two subpanels of panel F, Can the authors show how phase precession looks at these speeds in the experimental data and in their model, and compare it with the alternative model?

Please discuss how much of these behavioral-dependent sweeps could be thought of a side effect of a behavioral-dependent place field size and density resulting from the process of place cells learning their place fields.

---

## [Author Response]

Essential revisions:1. The authors conclude the experimental data best fit the behavior-dependent sweep model. However, the predicted effects on theta trajectory length and place cell properties are very similar between the temporal sweep model and the behavior-dependent sweep model (Figure 7). Therefore, it is unclear if they can conclusively determine experimental data best fit the behavior-dependent sweep model rather than the temporal sweep model, especially based on visual inspection. Their argument would be strengthened by analytically directly comparing the models and experimental data to determine which is a best fit. For example, is there are larger different between predicted and real data for the spatial and temporal sweep models than the behavior-dependent sweep model? A deeper analysis of what specifically differentiates the temporal sweep and behavior-dependent sweep is warranted. For example, the authors state "The temporal sweep model could be fitted to roughly capture the observed increase in theta trajectory lengths, place field sizes and phase precession slopes pooled by running speed, but it incorrectly predicted changes in individual fields with speed." Visually these "incorrect predictions" do not seem that far off. Are they significantly different and if so what is the magnitude of the difference?

We agree with this point and we have tried to improve on it. First, we have made the visualization in Figure 7 clearer. Initially, the within-field increases in place field sizes and phase precession slopes corresponded to the increases between consecutive, partly overlapping, speed bins. This was suboptimal, since it did not highlight potential cumulative changes in place field sizes and phase precession slopes across speed bins. As a result, it under-represented the magnitude of the discrepancy between the data and temporal sweeps. Therefore, we have now plotted within-field increases in place field sizes and phase precession slopes as a function of the deviation from the field’s characteristic running speed. This visualization shows that individual cells get increasingly larger fields and shallower phase precession slopes with speed in the temporal sweep model, but not in the experimental data or in the behavior-dependent sweep model.

Second, we have simplified the color code in the figure to reflect only whether the model fits with the experimental data qualitatively or not (e.g., whether a variable that increases with speed in the experimental data also does so in the simulated data). The qualitative differences are now all pronounced and it is clear that only the behavior-dependent sweep qualitatively fits with all of the results. Third, we have provided quantitative measures for the fit of the models to the experimental data. In particular, we report the mean squared error (difference) between the average values produced by the model and the experimental data for each speed bin. The behavior-dependent sweep model presents the lowest errors for 4 of the 5 analyzed variables.

2. The behavior-dependent sweep model (equation 3) is essentially the spatial sweep model with dependence on the averaged speed which is a function of location. There is no instantaneous speed term in this model, even though they show the effects of instantaneous running speed in the model (Figure 2). The authors do not describe how exactly variation in instantaneous speed affect the model. Does the model only use average speed and if so, over what time scales? If the model uses average speed over long timescales like a whole trial or multiple trials, how does the model account for rapid changes in speed levels, like at the scale of a single theta cycle?

We hypothesize that in the behavior-dependent sweep model, the characteristic speed is computed over a sufficiently long time scale that filters out variation in running speeds from run to run. In fact, in our simulations, we used the average speed in a given session (excluding outliers). However, we do not commit to any particular view regarding how exactly this characteristic speed is calculated in the brain (e.g., it could be some kind of moving average (cumulative, exponential, etc.), or it could be set in the first few runs and then remain mostly stable). We have now clarified this issue where the term is first introduced, and we also discuss some possibilities in the section “Potential mechanisms of behavior-dependent sweeps”. Future experiments are needed to determine the exact nature and timescale of this average. Due to the hypothesized setting of the characteristic speed on a long timescale, it is not affected by rapid changes in running speed (within a trial or a single theta cycle). Therefore, in the behavior-dependent sweep model, single cell variables that depend on characteristic speed, such as place field size and phase precession slope do not change with instantaneous speed either (Figures 4 and 5).

However, in behavior-dependent sweeps, as well as in spatial sweeps, theta trajectory lengths are still somewhat affected by instantaneous running speed. This is because the sweep starts at some fixed distance behind the animal at the beginning of the theta cycle (t_start_), and ends at some fixed distance ahead of the animal at the end of the theta cycle (t_end_). However, the current position of the animal, x(t), changes slightly during that time (t_end_-t_start_ = T) due to the animal’s motion. This change depends on the instantaneous running speed v and is given by x(t_end_)-x(t_start_) ≈ v*T to a first approximation. This distance v*T is added to the constant d_theta_, the sum of look-behind and look-ahead distances, to obtain the total theta trajectory length. We had described this effect for the spatial sweep model, but failed to alert the reader to the same effect arising in the behavior-dependent model (which is illustrated in Figure 2). We rectified this oversight.

Relatedly, the use of the speed term needs to be clarified in several places:a. In Figure 7D, cell firing properties are separated for different speed levels. Over what time scales was speed measured? Are these speed levels the "average speed" term in equation (3)?

We used the label “running speed” (without further qualifiers) consistently to refer to the instantaneous speed because we felt that the former was more standard even though the latter is more precise. Thus, “running speed” in Figure 7D refers to instantaneous running speed as in the other panels of Figure 7 and in Figures 3-5. The characteristic speed term in Eqation 3 was used when generating spiking activity from the behavior-dependent model, but this activity is then analyzed and plotted as a function of instantaneous speed in Figure 7D. The only time we show the characteristic speed in a figure is Figure 6, where we explicitly use the label “characteristic speed“. We made our use of terms explicit.

b. Based on equation (21), phase precession slope depends on the average speed in the behavior-dependent sweep model. How does this fit on single trial phase precession slope? For single phase precession, is the "averaged speed" calculated from the limited data within that trial?

This point is probably already clarified by our replies above. The characteristic speed that controls phase precession slopes is calculated over multiple runs through the environment. The average speed of a single place field traversal is in fact what we take to be the traversal’s instantaneous running speed. This is because we select place field traversals during which rats ran at approximately constant speed. We show that this traversal speed does not affect phase precession slopes (Figure 5 D, E). In addition to clarifying what we mean by “characteristic running speed”, we have added a note in the Methods reminding the reader that the average speed of a single place field traversal is different from what we refer to as characteristic running speed.

3. Related to the above points, the two models that the authors contrast with their own with are oversimplified. Rather than highlighting the virtues of their own model, this creates the impression that any model in which place cells do not have unrealistic properties, such as place field width vanishing as the animal slows down, would work equally well. This impression is reinforced by the fact that crucial aspects of the model are not tested, and instead very general properties, some of which were already known, are tested as predictions of the model: running speed and properties of place cells vary along the track. To increase the impact of this work, the text should be revised to ensure that the authors' model is not compared with overly simplified models.

We would like to address this complex point in three parts. A: We argue that while the spatial and temporal sweep models are simple they are not unrealistic. B: Whether one considers them oversimplified or not, we believe there are important reasons to include them in our study since the ideas underlying spatial and temporal sweeps have played an important role in the literature and shaped our implicit assumptions about theta sequences. C: We agree that one could test for more aspects of the model and we have improved on this point.

A.1 The reviewers seem concerned that in the temporal sweep model, place field sizes vanish as the animal slows down. This is actually not the case. We have now tried to clarify this point in the manuscript. In the temporal sweep model, at zero speed, the position represented by the population, r(t), remains fixed at the animal’s current position. If a cell only fired when r(t) crossed the cell’s preferred position, defined as a single point in space, then the cell’s place field would become infinitely narrow as speed goes to zero. However, the cell’s preferred position is not defined as a point but as a ‘true’ place field with a finite non-negligible size (more on the assumption of true place field sizes in our reply to comment 6). In our simulations of the spatial and temporal sweep models, the true place field size is ~30cm. Therefore, cells fire everywhere within the cells’ true place fields, even at zero speeds, which acts as a lower bound on the measured place field size. Thanks to this feature, the temporal sweep model does not do a particularly bad job at reproducing experimentally observed place field sizes (Figure 7B). We also note that the reviewers themselves have pointed out above that the advantage of behavior-dependent sweeps over temporal sweeps was not evident enough.

A.2 We note that the simulations of the temporal sweep (Figure 7B) and spatial sweep (Figure 7C) models were performed using experimentally measured theta phases and position data in the same way as they were for the behavior-dependent model (Figure 7D). So, the three models are compared on an equal footing.

A.3 The behavior-dependent sweep model is only minimally more complex than the other two. As discussed in the manuscript and above in comment 2, the behavior-dependent sweep model is like the spatial sweep model where place field size depends on characteristic speed (Equation 5). Alternatively, it can also be seen as an approximation of temporal sweeps where characteristic speed is used instead of instantaneous speed. We feel that this additional assumption does not put the behavior-dependent sweep model in a different class of models from the other two.

B.1 The spatial and temporal sweep models are the most straightforward explanation for certain subsets of experimental findings, and as such deserve attention. The spatial sweep model is the simplest model that explains constant place field sizes and phase precession slopes, and temporal sweeps are the simplest explanation for the approximately proportional increase in theta trajectory length with speed observed by (Maurer et al., 2012) and (Gupta et al., 2012).

B.2 The models serve the function of articulating two ways of understanding the theta phase code that are more or less implicit in the literature. Spatial and temporal sweeps crystallize the ideas that different phases of theta encode positions “behind or ahead” or “in the past and future” respectively, which are pervasive in the field. Most researchers likely use these terms rather loosely, and do not endorse spatial nor temporal sweeps fully. However, we believe it is important to highlight what this language implies when taken literally, as this can help bring greater clarity and understanding.

B.3 Sometimes this language is, in fact, used literally or reflected explicitly in computational models. For instance, (Itskov et al., 2008) write that “place cell activity on different phases of theta reflects positions shifted into the future or past along the animal’s trajectory in a two-dimensional environment”. They conclude this after they find that among several models, the one that best fits experimental data is essentially what we describe as the temporal sweep (with the only difference that r(t) does not jump abruptly between theta cycles as in our model). (Chadwick et al., 2015)’s model of phase precession and theta trajectories, on the contrary, assumes that the phase-vs-position relationship of all cells is identical, consistent with what we describe as spatial sweeps. Most computational modeling work using place cells for path integration or goal directed navigation (e.g., (Brzosko et al., 2017; Foster et al., 2000; Samsonovich and McNaughton, 1997)) also make the simplifying assumption that all place fields are identical, which is again consistent with spatial sweeps.

B.4 Spatial and temporal sweeps set the context for understanding behavior-dependent sweeps, which can be construed as combining aspects of both.

C.1 When introducing the behavior-dependent sweep model, we have added equations showing how theta trajectory lengths and place field sizes depend linearly, and phase precession slopes hyperbolically, on characteristic running speed. This makes it easier to understand why the data shown on Figure 6 supports the behavior-dependent sweep model.

C.2 Since the model predicts a hyperbolic relationship between phase precession slopes and characteristic running speed, plotting the inverse of phase precession slopes should reveal a linear relationship. Therefore in Figure 6 we have plotted inverse phase precession slopes instead of phase precession slopes.

C.3 We have added moving averages to the plots showing place field size and inverse phase precession slope vs characteristic speed. The moving averages shows that the relationship between inverse phase precession slope and characteristic speed is indeed approximately linear for almost the complete range of speeds. The increases in theta trajectory lengths and place field sizes with characteristic speed also appear very linear for up to ~60 cm/s, which encompasses most of the data, and then flatten out. In the new supplementary figures S4 we show that the relationship between theta trajectory length and characteristic speed for each individual animal does not seem to flatten out, suggesting that the flattening in the pooled data is at least partly caused by combining points from clouds with different slopes and speed ranges.

C.4 We discuss whether we could additionally compare behavior-dependent sweeps to other models in our reply to comment 5.

4. The authors should make sure that prior work from other labs is presented in its proper context and not specifically associated with one of the two overly simplified models presented in the text.

We did not intend to imply that our colleagues have explicitly endorsed or would endorse either of these models. We simply meant to point out which individual previous findings were consistent with each of these conceptual models in the regimes that were examined. We have clarified this where possible.

There might have also been some specific misunderstandings regarding our citations of (Maurer et al., 2012) and (Geisler et al., 2007). Upon reflection we realized that we used the reference to Maurer et al., in a confusing way, since we cite them as evidence for both spatial and temporal sweeps without clarification. Their main findings (theta trajectory length increasing proportionally with speed) appear consistent with temporal sweeps, whereas their supplementary figure 1 (place field sizes do not change with speed) appears consistent with spatial sweeps. We clarified which result of Maurer et al., is consistent with which model. We also note that we have discussed our work with Andrew Maurer, who does not feel we have misrepresented his findings or interpretations.

We cited the work by Geisler et al., in support of spatial sweeps, while Reviewer 3 comments that Geisler et al., seem to claim the opposite, for example based on the following quote: 'Furthermore, the phase advance of the spikes relative to theta was steeper in fast than in slow trials'. The key to resolving this apparent contradiction is the underlined phrase. Geisler et al., describes phase precession relative to theta. We refer to phase precession with respect to position. There is actually no contradiction. In order to keep the phase vs. position relationship fixed (as described by spatial sweeps), the phase vs. theta/time relationship needs to become steeper with increasing speed. By contrast, if the phase vs. theta/time relationship remained constant, it would always take the same number of theta cycles to complete the phase precession, and that would result in place field sizes and (spatial) phase precession slopes increasing with speed. Hence, Geisler et al., conclude: “Here we directly show that the temporal-phase precession slope, and therefore the oscillation frequency of place cells, is positively correlated with the locomotion speed of the rat, such that the phase-distance relationship remains invariant”. We clarified in our manuscript that phase precession can be/was studied with respect to different variables.

5. To address the issue whether field size or speed is better correlated with theta (spatial) length, the authors may consider producing a plot similar to Figure 7A, but using (mean) place field size rather than (mean) running speed, and adding a simulation condition using mean place field size as a controlling variable. The outcome could be one is better than the other, or they are similar or equivalent. Regardless of the outcome, the authors could discuss further the functional differences between these scenarios.

Following the reviewers' request, we added a new supplementary figure S3 which shows the mutual relationships between theta trajectory lengths, place field sizes and also (inverse) phase precession slopes. Based on the equations we derive in the section “Mathematical description of coding schemes”, we expected these variables to display linear relationships to one another. This is indeed the case. The three variables display noisy but strikingly linear relationships to each other. The relationships between theta trajectory length and either mean place field size or mean (inverse) phase precession slope have correlation coefficients of 0.29, whereas the relationship between theta trajectory length and characteristic running speed has a slightly stronger correlation coefficient of 0.35. A couple of explanations for this small discrepancy quickly come to mind. First, some cells participating in the analysis of theta trajectory lengths were excluded from the analysis of place field size or phase precession slope because they displayed overlapping phase precession clouds. Second, place fields are skewed in systematic ways (Figure S6A). Theta trajectory lengths will be shorter or longer depending on whether they occur on the shorter or longer tails of the place fields. The place field size is calculated over the whole field and therefore misses this effect. Based on these results and considerations, we would expect a model that takes place field sizes as input to perform reasonably well, especially if it can also take into account other measures such as place field skews. However, note that theta trajectory lengths, place field sizes, skews or phase precession slopes are all partly redundant measurements of the same underlying neuronal activity. As such, it must be possible to reconstruct one measurement out of combinations of the others. But this would offer no direct insight as to why any of the variables changes in the first place. The three models we have put forward and contrasted to each other attempt to provide an account for the theta phase code. But what the cells are coding for must necessarily be described in terms of variables that are external to the place cells themselves. This is why we decided not to include results on a model where d_theta depends on the width and density of place fields or some other combination of place cell measurements.

6. What is the evidence for the assumption of a "true place field"? What are the potential limitations of this assumption?

The idea of a ‘true’ place field dates back at least to (Tsodyks et al., 1996)’s model of phase precession and virtually all experimental studies on theta sequences assume (even if implicitly) that the true place field of a place cell differs from its measured firing field. In the Tsodyks et al., model, at the beginning of each theta cycle, the place cells selective for the current position of the animal become activated, and then trigger the activation of cells that represent positions progressively further ahead. This results in the fact that only firing in the last part of the measured place field (where cells fire at the beginning of the theta cycle) is indicative of the animal being at the cells’s preferred location. John Lisman explicitly used the term “true place field” to refer to this preferred firing location when discussing a similar model (Lisman et al., 2005; Sanders et al., 2015). In experimental studies, it is generally agreed upon that the position represented by the hippocampal population sweeps forward during theta cycles, possibly starting behind the animal and reaching ahead (e.g. (Cei et al., 2014; Dragoi and Buzsáki, 2006; Drieu and Zugaro, 2019; Gupta et al., 2012; Itskov et al., 2008; Jensen and Lisman, 1996; Kay et al., 2020; Muessig et al., 2019; Terada et al., 2017)). This actually implies the existence of a ‘true’ place field. The true place field is, by definition, the portion of the field where the cell fires when representing the current position of the animal. When cells fire to represent positions behind or ahead of the animal, they would be firing outside of their true place fields. Another way of looking at it is that if the firing of place cells is indicative of a represented location r(t), which deviates from the current location x(t), then cells can fire when r(t) is located inside the cell’s preferred location while x(t) is not. That would result in firing outside of the cell’s preferred location or true place field. The opposite can also happen: that a place cell fails to spike when the animal is located within its true place field because the population is representing an r(t) that lies outside of it.

The alternative to the concept of true place fields would be to assume that cells always represent the current position of the animal (r(t) is always equal to x(t)). In this view, the change in theta phase across the field would just add more precision to the spatial code (e.g., cell A firing at 300º indicates that the animal is currently at position x, and the same cell firing at 60º indicates that the animal is currently at a different position y) (O’Keefe and Recce, 1993). However, in 2D open fields, the one-to-one mapping between positions and phases breaks down. A cell will fire at different phases in the same position depending on whether it is arriving at the field or exiting it (Huxter et al., 2008; Jeewajee et al., 2014). Hence, the spatial code would become less straightforward, having to take into account both theta phase and movement direction to decode the current spatial position. The interpretation that firing at different phases reflects arriving at or coming from the cell’s true place field then becomes more parsimonious. It also seems to fit better with accounts of phase precession reversal during backwards travel (Cei et al., 2014; Maurer et al., 2014), prospective shifts in the location of place fields (Battaglia et al., 2004), or constant sub-second cycling between representations of possible futures (Kay et al., 2020), as well as with the potential role of theta phase precession in memory and planning (Bolding et al., 2020; Robbe and Buzsáki, 2009; Zheng et al., 2021).

So the assumption of true place fields, although rarely made explicit, is actually deeply ingrained in the field. If it turned out to be wrong, a reinterpretation of most of the literature on theta phase coding would be required and some common methods would need to be re-evaluated (e.g., the ways in which position is decoded from ensemble place cell activity would turn out be misleading, since they produce theta trajectories that constantly deviate from the current position of the animal).

We changed the text to discuss the concept of true place fields explicitly in the manuscript.

7. To increase impact, the overall organization and presentation of ideas should be revised to ensure clarity for a general audience.

After making numerous clarifications and improvements throughout the text motivated by the reviewer’s comments, we feel that the manuscript has become much more accessible for a general audience.

We considered trying to resolve the contradictions earlier, in Figures 4 and 5, as opposed to waiting until Figure 6, like Reviewer 3 suggested. We decided not to do so because we think it is helpful to keep a two-parts structure, where in the first part we reproduce in the same dataset the combination of conflicting results, and in the second part we offer a solution to the conflict. The solution in Figure 6 now includes 9 panels. We believe it is beneficial to consider all of them together when trying to understand the proposed solution, so we prefer to dedicate a figure for this purpose. However, we have tried to add more early hints as to what the solution will be so that readers can better navigate the manuscript.

Reviewer #3:The main result of the authors is a model that is compatible with experimental results. In my view, they should put some effort into clarifying whether this is a unique model or similar proposals would perform equally well. This means dissecting the crucial aspects that make this model better than any other (not just the two oversimplified models with unrealistic place field properties). Variations that I can think of are, for example, temporal sweeps with an overall mean speed rather than the mean speed for a given position, or spatial sweeps with a d_theta that depends on the width and density of place fields. If they find that there is a family of models sharing some crucial element that perform similarly, this would strengthen rather than weaken the manuscript.

It is a fair point to ask whether other models might account for the data equally well or better. However, the options advanced by the reviewer are problematic. Temporal sweeps with an overall mean speed rather than the mean speed at each position would end up being equivalent to spatial sweeps, because the look-ahead and look-behind distances are independent of location or characteristic speed. Spatial sweeps with a d_theta that depends on place field sizes would work, but as we explain in our reply to the essential revision number 5, we believe that such a model would play a different kind explanatory role, and is therefore not directly comparable to the models we consider.

In trying to improve on this point, we have tested another explanatory variable: the distance to the nearest end of the track, which is also the distance to the nearest reward location. This variable also turns out to have a significant effect on place field parameters, but characteristic running speed uniquely accounts for a larger fraction of the variance (when using linear models). Of course, it is possible that the relationship between place field parameters and distance to borders or reward is non-linear. Further experiments are needed to fully dissociate the two variables. These results were added to the Results.

Finally, in the Discussion section we have tried to provide more alternatives to behavior-dependent sweeps. For instance, we note that (Tanni et al., 2021) propose that place field sizes, and hence, presumably, theta trajectory lengths, depend on the spatial rate of change of visual input. Other alternatives could involve the density of sensory cues, the complexity of the maze, the level of retrospection and prospection required for the task, etc. Characteristic running speed is likely to be correlated with many of these variables. Further dedicated experimentation would be needed to determine which is more fundamental in driving the theta phase code.

30 – 'The current position of the animal'. There is an issue in how one defines this. What we typically have is an estimation of the position of LEDs that are placed somewhere above the animal's head. We ignore the distance between these LEDs and the position that CA1 considers to be the 'actual' position of the animal. Please comment on how this correction would affect the estimation of past and future positions at different speeds, especially when considering time sweeps.

This comment might be getting at two related issues. One is whether there is a shift between the position represented by CA1 and the position of the LEDs. This seems very likely. However, in a linear track, this would just mean that the position of all place fields and theta trajectories should be shifted forwards or backwards by some fixed amount. Other than that, place field and theta trajectory parameters should remain constant, including look-behind and look-ahead distances.

The second issue is whether we can know at which phase of theta the CA1 population represents the ‘actual’ position of the animal. This seems difficult, especially in a linear track. In our equations, the phase at which the current position is represented is θ_0_. This parameter determines the ratio between look-ahead and look-behind. As the reviewer points out, changing this ratio would have an effect on temporal sweeps in the presence of positive or negative acceleration. For instance, given positive acceleration, the more (temporal) look-ahead there is as compared to look-behind, the bigger the place fields would become. We have added this point to the discussion.

98 – This length does not seem to be affected by a potential increase in field size as a result of place cells increasing their firing rate with speed. Why is this not taken into account?

We assume that firing rate is modulated by a multiplicative factor that scales up or down the activity of place cells. This multiplicative scaling would mean that the shapes of place cells’ firing fields remain unaffected. Since the size of place fields is calculated by considering when the firing rate falls below a certain fraction of the peak firing rate, scaling the place fields would not affect their sizes. We observe an increase in firing rate with speed (Figure S6 B), but no change in place field sizes within individual cells (Figure 4B). Therefore, the assumption of multiplicative scaling seems reasonable. The multiplicative scaling would also mean that firing rate changes have no effect on theta trajectories. This is because the ratios between spike counts of different cells at different phases of theta, which drive the Bayesian decoding of position, would remain unaffected.

625 – Equation 12. It should be clarified that this is an approximation, since phase precession has been shown to be highly non-linear already by O'Keefe and Recce. Thus, the exact value of the slope is highly dependent on the method, especially on the relative sampling of different phase ranges.

We agree. We have added the following clarification:

“Note that the model makes the simplifying assumption that sweeps, and hence also phase precession, are linear. There are indications that this could be the case when analyzing single trials (Schmidt et al., 2009) or when calculating theta phase using a wave-form method (Belluscio et al., 2012) (but see (Souza and Tort, 2017; Yamaguchi et al., 2002)).”

108 – L = v(t). As I understand it, dimensions do not match. The same happens to Equation 16 in the methods. Do parenthesis stand for multiplication or evaluation of the function v at a given time?

Unfortunately, standard mathematical notation uses parenthesis for both multiplication and function evaluation. Here, they were meant to indicate multiplication, and therefore the units on both sides of the equation match. To avoid the ambiguity, we swapped the terms in the product, i.e., L=(tau+T)v.

108 – 'This is almost exactly the result reported by Maurer et al., (2012); Gupta et al., (2012).'. Please specify how and in which figures. For example, Figure 8 from Gupta et al., seems to be at odds with this claim.

Unfortunately, standard mathematical notation uses parenthesis for both multiplication and function evaluation. Here, they were meant to indicate multiplication, and therefore the units on both sides of the equation match. To avoid the ambiguity, we swapped the terms in the product, i.e., L=(tau+T)v.

112 – Please specify or give an example of which procedures or methods can cause a contradiction. It is not clear from the text.

We have added some preliminary suggestions here and now refer the reader to the Discussion, where discuss specific hypotheses in the subsection “Relationship to previous studies on the speed-dependence of the hippocampal code”. We feel that the details given there would be too much in the beginning of the paper.

Figure 4a – The visualization of these examples is really poor. Since the purpose of this panel is visualization, I suggest to find a better way, perhaps with fewer examples or speed bins.

We have zoomed in around the fields, which seems to help.

Figure 4b-c – Although the conclusion is plausible, and the examples seem to indicate this, the analysis is not convincing. The procedure based on regressing 7 datapoints has too much error, and this might mask an effect at the population level. (For example, are these individual regressions significant?) A high p-value does not necessarily mean that there is no effect. I suggest trying other methods. For example, the half-width decay in the spatial autocorrelation for the population of all cells could be studied and compared at different speeds. Fewer speed groups would also help getting better statistics. All of the analyses could also be applied to pools of cells with neighboring fields, to explain the contradiction right a way and thus help the reader navigate the manuscript.

We agree that regressing 7 datapoints (at most; there can be as few as two, depending on the number of speed bins with valid measurements) is definitely noisy and we do not expect those individual regressions to be statistically significant. However, we do not see any reason to suspect that the noise could be systematically biased in any particular direction (e.g., that the regressions would tend to produce positive or negative slopes in the absence of an underlying positive or negative relationship, respectively). Therefore, it is appropriate to aggregate many of those noisy measurements and then apply a statistical test to assess whether the population of place fields as a whole increases or decreases place field size or phase precession slope with speed. The p-values in Figure 4B refer to the population level test. The same principle is applied in all statistical tests, noisy individual measurements are pooled and tested at the population level, e.g. whether a drug alleviates an illness in a specific person is a very noisy measurement, but nevertheless one can find evidence for an effect in a large population.

With regards to the alternative analysis suggested, one could in principle compare the spatial autocorrelation for the population of cells at different speeds. However, that is a population measure and not a single-cell measure, which is what we focus on in Figure 4B. Nevertheless, since individual place field sizes do not change with speed, the half-width decay of the spatial autocorrelation of the population should not change either (we imagine this could be the reviewer’s reasoning). There is one problem with this reasoning: Not all positions were traversed at all speeds, e.g., some positions near the borders were never traversed at high speed; some positions near the center were never traversed at low speeds. Hence firing rate maps are undefined at some positions and speeds and would not participate in the autocorrelation. The autocorrelation would then reflect data from different positions at different speeds. In other words, the population vector cannot be used in this way to study within-cell changes due to speed. In the end, we would see a similar effect of speed on autocorrelation width as we seen when we combine place field sizes across fields for each speed bin (Figure 4C). That relationship is statistically significant.

178 – It is not clear at this point of the manuscript why both results are compatible with behavior dependent sweeps. This might need some more explanation.

We now make that more explicit that this is the prediction depicted in Figure 2.

191 – I suggest to try to make explicit and consistent the description of slopes. Unfortunately, describing a negative slope as 'lower' is ambiguous. Similarly, in 209 an 'increase in the slope' is mentioned, which is also ambiguous. Please revise this language throughout the manuscript. I believe people in the past have referred to slopes as shallower or steeper to avoid ambiguity.

We agree with the reviewer and have made the suggested change.

181 – 194. What would the different models predict and why?

After the single pass phase precession analysis we briefly commented on which of the results fitted which of the models. This was meant to apply also to the pooled phase precession results discussed above, but that was not clear. We have improved on this and also added references to the relevant equations.

234 – Why are these results striking? Similar findings have been reported before. For example in Gupta et al., 2012. I disagree with lines 312 – 314. Figure S3 of Gupta et al. shows very similar results to the ones analyzed here.

That is a fair point. We removed the word “striking” and changed “Two previous studies (Maurer et al., 2012; Gupta et al., 2012) pooled theta trajectories based exclusively on speed, therefore masking the role of position in mediating the relationship between speed and theta trajectory length” to “[…] pooled theta trajectories across different positions when showing the effect of speed on theta trajectory length”.

Figure 6A – If this explains the apparent contradictions found in Figures 4 and 5, then pooling locations with similar average speed should make the pooled average effects disappear. This needs to be shown in those figures to help readers navigate the manuscript.

The reviewer is correct in principle, however, the suggested analysis is unlikely to add more clarity. If the range of characteristic speeds considered was too narrow, too few place fields would be pooled together, and the relationship between place field size and speed would be very noisy. If, on the other hand, a bigger range of characteristic speeds was used to pool place fields, we would start to see again the problem that different place fields contribute to the low and high speed averages. However, we have added some more early hints as to what is happening so that the readers are less confused in the build-up to Figure 6.

Figure 6,B-C – running speed at 0.25 is leeds then two thirds of the speed at 0.75. However theta length trajectories are similar. To my view this speaks directly against the simple explanation that is being tested 'Short theta trajectories and small place field sizes could occur where animals tend to run slow, contributing to the low speed averages, and long theta trajectories and large place field sizes could occur where animals tend to run fast, contributing to the fast speed averages'. There seems to be something else going on here.

The reviewer is right, this a curious effect that might arise due to interindividual differences between animals or differences in acceleration at different locations. Regarding the first point, the relationships between theta trajectory length and characteristic speed for each animal are mostly linear (new Figure S4), whereas the relationship flattens in the pooled data (new Figure 6G). We added a discussion of this point to the Results. Regarding the second point, there might be differences in acceleration between the first and second halves of the track. If there was more look-ahead than look-behind, then, given equal speed, theta trajectories could be longer in areas of positive acceleration. As we had commented in the Discussion, future work is needed to the study the potential effects of acceleration for future work where the data analyzed contains periods of more sustained acceleration and deceleration.

329 – Figure S5. In this paragraph an alternative model is discarded, but no quantification is provided. In the last two subpanels of panel F, Can the authors show how phase precession looks at these speeds in the experimental data and in their model, and compare it with the alternative model?

To facilitate a fairer evaluation of the Chadwick et al., model, we now quantify the differences between the experimental data and our “variable noise model”, which is based on the same principle as Chadwick et al.,’s, but was fitted to our experimental data. The conclusion remains that this type of model cannot account for some key aspects in our data.

Please discuss how much of these behavioral-dependent sweeps could be thought of a side effect of a behavioral-dependent place field size and density resulting from the process of place cells learning their place fields.

As we discussed in the reply to Essential Revisions no. 5, since place field size is correlated with characteristic speed, we cannot rule out that the effect of characteristic speed on theta coding is mediated via place field size. We have therefore added the following to the section titled “Potential mechanisms of behavior-dependent sweeps”:

“Alternatively, on the view that theta sequences arise from independent single cell coding (Chadwick et al., 2015), behavior-dependent sweeps could be seen as resulting from larger learned place fields in areas of faster running.”

References

Battaglia, F. P., Sutherland, G. R., and McNaughton, B. L. (2004). Local sensory cues and place cell directionality: Additional evidence of prospective coding in the hippocampus. *The Journal of Neuroscience : The Official Journal of the Society for Neuroscience*, *24*(19), 4541–4550. https://doi.org/10.1523/JNEUROSCI.4896-03.2004

Belluscio, M. A., Mizuseki, K., Schmidt, R., Kempter, R., and Buzsáki, G. (2012). Cross-Frequency Phase–Phase Coupling between Theta and Γ Oscillations in the Hippocampus. *The Journal of Neuroscience*, *32*(2), 423–435. https://doi.org/10.1523/jneurosci.4122-11.2012

Bolding, K. A., Ferbinteanu, J., Fox, S. E., and Muller, R. U. (2020). Place cell firing cannot support navigation without intact septal circuits. *Hippocampus*, *30*(3), 175–191. https://doi.org/10.1002/hipo.23136

Brzosko, Z., Zannone, S., Schultz, W., Clopath, C., and Paulsen, O. (2017). Sequential neuromodulation of Hebbian plasticity offers mechanism for effective reward-based navigation. *eLife*, *6*, e27756. https://doi.org/10.7554/*eLife*.27756

Cei, A., Girardeau, G., Drieu, C., Kanbi, K. E., and Zugaro, M. (2014). Reversed theta sequences of hippocampal cell assemblies during backward travel. *Nat Neurosci*, *17*(5), 719–724. https://doi.org/10.1038/nn.3698

Chadwick, A., van Rossum, M. C., and Nolan, M. F. (2015). Independent theta phase coding accounts for CA1 population sequences and enables flexible remapping. *ELife*, *4*, e03542. https://doi.org/10.7554/*eLife*.03542

Dragoi, G., and Buzsáki, G. (2006). Temporal Encoding of Place Sequences by Hippocampal Cell Assemblies. *Neuron*, *50*(1), 145–157. https://doi.org/10.1016/j.neuron.2006.02.023

Drieu, C., and Zugaro, M. (2019). Hippocampal Sequences During Exploration: Mechanisms and Functions. *Frontiers in Cellular Neuroscience*, *13*. https://doi.org/10.3389/fncel.2019.00232

Foster, D. J., Morris, R. G., and Dayan, P. (2000). A model of hippocampally dependent navigation, using the temporal difference learning rule. *Hippocampus*, *10*(1), 1–16. https://doi.org/10.1002/(SICI)1098-1063(2000)10:1<1::AID-HIPO1>3.0.CO;2-1

Geisler, C., Robbe, D., Zugaro, M., Sirota, A., and Buzsáki, G. (2007). Hippocampal place cell assemblies are speed-controlled oscillators. *Proceedings of the National Academy of Sciences of the United States of America*, *104*(19), 8149–8154. https://doi.org/10.1073/pnas.0610121104

Gupta, A. S., van der Meer, M. A., Touretzky, D. S., and Redish, A. D. (2012). Segmentation of spatial experience by hippocampal theta sequences. *Nature Neuroscience*, *15*(7), 1032–1039. https://doi.org/10.1038/nn.3138

Huxter, J. R., Senior, T. J., Allen, K., and Csicsvari, J. (2008). Theta phase-specific codes for two-dimensional position, trajectory and heading in the hippocampus. *Nature Neuroscience*, *11*(5), 587–594. https://doi.org/10.1038/nn.2106

Itskov, V., Pastalkova, E., Mizuseki, K., Buzsaki, G., and Harris, K. D. (2008). Theta-mediated dynamics of spatial information in hippocampus. *The Journal of Neuroscience*, *28*(23), 5959–5964. https://doi.org/10.1523/JNEUROSCI.5262-07.2008

Jeewajee, A., Barry, C., Douchamps, V., Manson, D., Lever, C., and Burgess, N. (2014). Theta phase precession of grid and place cell firing in open environments. *Philosophical Transactions of the Royal Society B: Biological Sciences*, *369*(1635), 20120532. https://doi.org/10.1098/rstb.2012.0532

Jensen, O., and Lisman, J. E. (1996). Hippocampal CA3 region predicts memory sequences: Accounting for the phase precession of place cells. *Learning and Memory (Cold Spring Harbor, N.Y.)*, *3*(2–3), 279–287.

Kay, K., Chung, J. E., Sosa, M., Schor, J. S., Karlsson, M. P., Larkin, M. C., Liu, D. F., and Frank, L. M. (2020). Constant Sub-second Cycling between Representations of Possible Futures in the Hippocampus. *Cell*, *180*(3), 552-567.e25. https://doi.org/10.1016/j.cell.2020.01.014

Lisman, J. E., Talamini, L. M., and Raffone, A. (2005). Recall of memory sequences by interaction of the dentate and CA3: A revised model of the phase precession. *Neural Networks*, *18*(9), 1191–1201. https://doi.org/10.1016/j.neunet.2005.08.008

Maurer, A. P., Burke, S. N., Lipa, P., Skaggs, W. E., and Barnes, C. A. (2012). Greater running speeds result in altered hippocampal phase sequence dynamics. *Hippocampus*, *22*(4), 737–747.

Maurer, A. P., Lester, A. W., Burke, S. N., Ferng, J. J., and Barnes, C. A. (2014). Back to the future: Preserved hippocampal network activity during reverse ambulation. *The Journal of Neuroscience : The Official Journal of the Society for Neuroscience*, *34*(45), 15022–15031. https://doi.org/10.1523/jneurosci.1129-14.2014

Muessig, L., Lasek, M., Varsavsky, I., Cacucci, F., and Wills, T. J. (2019). Coordinated Emergence of Hippocampal Replay and Theta Sequences during Post-natal Development. *Current Biology*, *29*(5), 834-840.e4. https://doi.org/10.1016/j.cub.2019.01.005

O’Keefe, J., and Recce, M. L. (1993). Phase relationship between hippocampal place units and the EEG theta rhythm. *Hippocampus*, *3*(3), 317–330. https://doi.org/10.1002/hipo.450030307

Robbe, D., and Buzsáki, G. (2009). Alteration of theta timescale dynamics of hippocampal place cells by a cannabinoid is associated with memory impairment. *The Journal of Neuroscience : The Official Journal of the Society for Neuroscience*, *29*(40), 12597–12605. https://doi.org/10.1523/JNEUROSCI.2407-09.2009

Samsonovich, A., and McNaughton, B. L. (1997). Path integration and cognitive mapping in a continuous attractor neural network model. *The Journal of Neuroscience*, *17*(15), 5900–5920.

Sanders, H., Rennó-Costa, C., Idiart, M., and Lisman, J. (2015). Grid Cells and Place Cells: An Integrated View of their Navigational and Memory Function. *Trends in Neurosciences*, *38*(12), 763–775. https://doi.org/10.1016/j.tins.2015.10.004

Schmidt, R., Diba, K., Leibold, C., Schmitz, D., Buzsáki, G., and Kempter, R. (2009). Single-trial phase precession in the hippocampus. *The Journal of Neuroscience : The Official Journal of the Society for Neuroscience*, *29*(42), 13232–13241. https://doi.org/10.1523/jneurosci.2270-09.2009

Souza, B. C., and Tort, A. B. L. (2017). Asymmetry of the temporal code for space by hippocampal place cells. *Scientific Reports*, *7*(1), 8507. https://doi.org/10.1038/s41598-017-08609-3

Tanni, S., Cothi, W. de, and Barry, C. (2021). State transitions in the statistically stable place cell population are determined by visual change. *BioRxiv*, 2021.06.16.448638. https://doi.org/10.1101/2021.06.16.448638

Terada, S., Sakurai, Y., Nakahara, H., and Fujisawa, S. (2017). Temporal and Rate Coding for Discrete Event Sequences in the Hippocampus. *Neuron*, *94*(6), 1248-1262.e4. https://doi.org/10.1016/j.neuron.2017.05.024

Tsodyks, M. V., Skaggs, W. E., Sejnowski, T. J., and McNaughton, B. L. (1996). Population dynamics and theta rhythm phase precession of hippocampal place cell firing: A spiking neuron model. *Hippocampus*, *6*(3), 271–280. https://doi.org/10.1002/(sici)1098-1063(1996)6:3\%3C271::aid-hipo5\%3E3.0.co;2-q

Yamaguchi, Y., Aota, Y., McNaughton, B. L., and Lipa, P. (2002). Bimodality of theta phase precession in hippocampal place cells in freely running rats. *J Neurophysiol*, *87*(6), 2629–2642.

Zheng, C., Hwaun, E., Loza, C. A., and Colgin, L. L. (2021). Hippocampal place cell sequences differ during correct and error trials in a spatial memory task. *Nature Communications*, *12*(1), 3373. https://doi.org/10.1038/s41467-021-23765-x